# FROM $t$-SNE TO UMAP
# WITH CONTRASTIVE LEARNING

**Sebastian Damrich**
IWR at Heidelberg University
sebastian.damrich@uni-tuebingen.de

**Jan Niklas Böhm**
University of Tübingen
jan-niklas.boehm@uni-tuebingen.de

**Fred A. Hamprecht**
IWR at Heidelberg University
fred.hamprecht@iwr.uni-heidelberg.de

**Dmitry Kobak**
University of Tübingen
dmitry.kobak@uni-tuebingen.de

## ABSTRACT

Neighbor embedding methods $t$-SNE and UMAP are the de facto standard for visualizing high-dimensional datasets. Motivated from entirely different viewpoints, their loss functions appear to be unrelated. In practice, they yield strongly differing embeddings and can suggest conflicting interpretations of the same data. The fundamental reasons for this and, more generally, the exact relationship between $t$-SNE and UMAP have remained unclear. In this work, we uncover their conceptual connection via a new insight into contrastive learning methods. Noise-contrastive estimation can be used to optimize $t$-SNE, while UMAP relies on negative sampling, another contrastive method. We find the precise relationship between these two contrastive methods and provide a mathematical characterization of the distortion introduced by negative sampling. Visually, this distortion results in UMAP generating more compact embeddings with tighter clusters compared to $t$-SNE. We exploit this new conceptual connection to propose and implement a generalization of negative sampling, allowing us to interpolate between (and even extrapolate beyond) $t$-SNE and UMAP and their respective embeddings. Moving along this spectrum of embeddings leads to a trade-off between discrete / local and continuous / global structures, mitigating the risk of over-interpreting ostensible features of any single embedding. We provide a PyTorch implementation.

## 1 INTRODUCTION

Low-dimensional visualization of high-dimensional data is a ubiquitous step in exploratory data analysis, and the toolbox of visualization methods has been rapidly growing in the last years (McInnes et al., 2018; Amid & Warmuth, 2019; Szubert et al., 2019; Wang et al., 2021). Since all of these methods necessarily distort the true data layout (Chari et al., 2021), it is beneficial to have various tools at one's disposal. But only equipped with a theoretic understanding of the aims of and relationships between different methods, can practitioners make informed decisions about which visualization to use for which purpose and how to interpret the results.

The state of the art for non-parametric, non-linear dimensionality reduction relies on the neighbor embedding framework (Hinton & Roweis, 2002). Its two most popular examples are $t$-SNE (van der Maaten & Hinton, 2008; van der Maaten, 2014) and UMAP (McInnes et al., 2018). Both can produce insightful, but qualitatively distinct embeddings. However, why their embeddings are different and what exactly is the conceptual relation between their loss functions, has remained elusive.

Here, we answer this question and thus explain the mathematical underpinnings of the relationship between $t$-SNE and UMAP. Our conceptual insight naturally suggests a spectrum of embedding methods complementary to that of Böhm et al. (2022), along which the focus of the visualization shifts from local to global structure (Fig. 1). On this spectrum, UMAP and $t$-SNE are simply two instances and inspecting various embeddings helps to guard against over-interpretation of apparent structure. As a practical corollary, our analysis identifies and remedies an instability in UMAP.

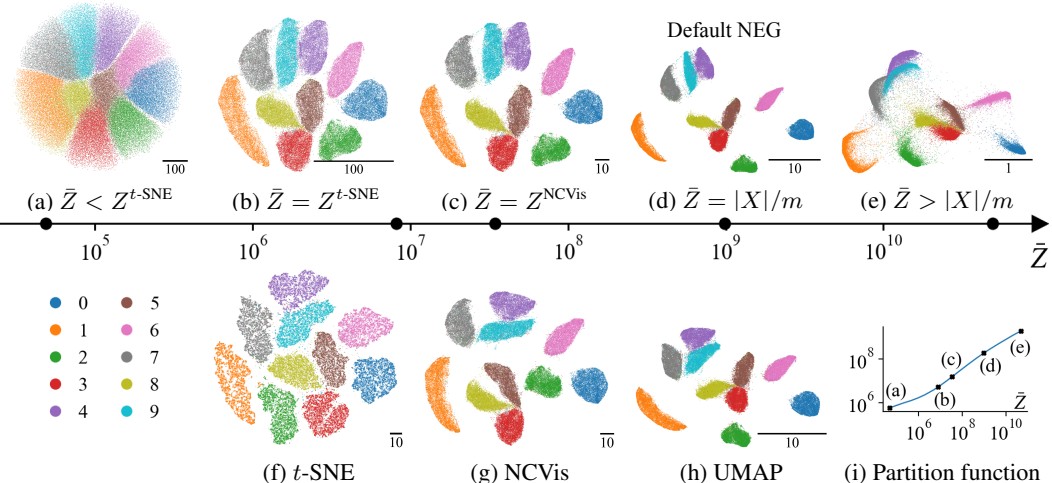

Figure 1: **(a–e)** Neg-$t$-SNE embedding spectrum of the MNIST dataset for various values of the fixed normalization constant $\bar{Z}$, see Sec. 5. As $\bar{Z}$ increases, the scale of the embedding decreases, clusters become more compact and separated before eventually starting to merge. The Neg-$t$-SNE spectrum produces embeddings very similar to those of **(f)** $t$-SNE, **(g)** NCVis, and **(h)** UMAP, when $\bar{Z}$ equals the partition function of $t$-SNE, the learned normalization parameter $Z$ of NCVis, or $|X|/m = \binom{n}{2}/m$ used by UMAP, as predicted in Sec. 4–6. **(i)** The partition function $\sum_{ij}(1+d_{ij}^2)^{-1}$ tries to match $\bar{Z}$ and grows with it. Here, we initialized all Neg-$t$-SNE runs using $\bar{Z} = |X|/m$; without this 'early exaggeration', low values of $\bar{Z}$ yield fragmented clusters (Fig. S11).

We provide the new connection between $t$-SNE and UMAP via a deeper understanding of contrastive learning methods. Noise-contrastive estimation (NCE) (Gutmann & Hyvärinen, 2010; 2012) can be used to optimize $t$-SNE (Artemenkov & Panov, 2020), while UMAP relies on another contrastive method, negative sampling (NEG) (Mikolov et al., 2013). We investigate the discrepancy between NCE and NEG, show that NEG introduces a distortion, and this distortion explains how UMAP and $t$-SNE embeddings differ. Finally, we discuss the relationship between neighbor embeddings and self-supervised learning (Wu et al., 2018; He et al., 2020; Chen et al., 2020; Le-Khac et al., 2020).

In summary, our contributions are

1. a new connection between the contrastive methods NCE and NEG (Sec. 4),
2. the exact relation of $t$-SNE and UMAP and a remedy for an instability in UMAP (Sec. 6),
3. a spectrum of 'contrastive' neighbor embeddings encompassing UMAP and $t$-SNE (Sec. 5),
4. a connection between neighbor embeddings and self-supervised learning (Sec. 7),
5. a unified PyTorch framework for contrastive (non-)parametric neighbor embedding methods.

Our code is available at `https://github.com/berenslab/contrastive-ne` and `https://github.com/hci-unihd/cl-tsne-umap`.

## 2 RELATED WORK

One of the most popular methods for data visualization is $t$-SNE (van der Maaten & Hinton, 2008; van der Maaten, 2014). Recently developed NCVis (Artemenkov & Panov, 2020) employs noise-contrastive estimation (Gutmann & Hyvärinen, 2010; 2012) to approximate $t$-SNE in a sampling-based way. Therefore, we will often refer to the NCVis algorithm as 'NC-$t$-SNE'. UMAP (McInnes et al., 2018) has matched $t$-SNE's popularity at least in computational biology (Becht et al., 2019) and uses another sampling-based optimization method, negative sampling (Mikolov et al., 2013), also employed by LargeVis (Tang et al., 2016). Other recent sampling-based visualization methods include TriMap (Amid & Warmuth, 2019) and PaCMAP (Wang et al., 2021).

Given their success, $t$-SNE and UMAP have been scrutinized to find out which aspects are essential to their performance. Initialization was found to be strongly influencing the global structure

in both methods (Kobak & Linderman, 2021). The exact choice of the low-dimensional similarity kernel (Böhm et al., 2022) or the weights of the $k$-nearest-neighbor graph (Damrich & Hamprecht, 2021) were shown to be largely inconsequential. Both algorithms have similar relevant hyperparameters such as, e.g., the heavy-tailedness of the similarity kernel (Yang et al., 2009; Kobak et al., 2019).

While not obvious from the original presentations, the central difference between $t$-SNE and UMAP can therefore only be in their loss functions, which have been studied by Damrich & Hamprecht (2021); Böhm et al. (2022); Wang et al. (2021), but never conceptually connected. We achieve this by deepening the link between negative sampling (NEG) and noise-contrastive estimation (NCE).

NEG was introduced as an ad hoc replacement for NCE in the context of learning word embeddings (Mikolov et al., 2013). The relationship between NEG and NCE has been discussed before (Dyer, 2014; Levy & Goldberg, 2014; Ruder, 2016; Ma & Collins, 2018; Le-Khac et al., 2020), but here we go further and provide the precise meaning of NEG: We show that, unlike NCE, NEG learns a model proportional but not equal to the true data distribution.

Both $t$-SNE and UMAP have parametric versions (van der Maaten, 2009; Sainburg et al., 2021) with very different logic and implementations. Here we present a unified PyTorch framework for non-parametric and parametric contrastive neighbor embeddings. As special cases, it includes UMAP and $t$-SNE approximations with NCE (like NCVis) and the InfoNCE loss (Jozefowicz et al., 2016; van den Oord et al., 2018), which has not yet been applied to neighbor embeddings.

Our resulting InfoNC-$t$-SNE elucidates the relationship between SimCLR (Chen et al., 2020) and neighbor embeddings. The parallel work of Hu et al. (2023) makes a qualitatively similar argument, but does not discuss negative samples. Balestriero & LeCun (2022) show how SimCLR recovers Isomap (Tenenbaum et al., 2000), while we connect SimCLR to more modern neighbor embeddings.

## 3 BACKGROUND

### 3.1 NOISE-CONTRASTIVE ESTIMATION (NCE)

The goal of parametric density estimation is to fit a parametric model $q_\theta$ to iid samples $s_1, \dots, s_N$ from an unknown data distribution $p$ over a space $X$. For maximum likelihood estimation (MLE) the parameters $\theta$ are chosen to maximize the log-likelihood of the observed samples

$$\theta^* = \arg\max_\theta \sum_{i=1}^N \log\left(q_\theta(s_i)\right). \tag{1}$$

This approach crucially requires $q_\theta$ to be a normalized model. It is otherwise trivial to increase the likelihood arbitrarily by scaling $q_\theta$. To circumvent the expensive computation of the partition function $Z(\theta) = \sum_{x \in X} q_\theta(x)$, Gutmann & Hyvärinen (2010; 2012) introduced NCE. It turns the unsupervised problem of density estimation into a supervised problem in which the data samples need to be identified from a set containing the $N$ data samples and $m$ times as many noise samples $t_1, \dots t_{mN}$ drawn from a noise distribution $\xi$, e.g., the uniform distribution over $X$. Briefly, NCE fits $\theta$ by minimizing the binary cross-entropy between the true class assignment and posterior probabilities $\mathbb{P}(\text{data} \mid x) = q_\theta(x)/\left(q_\theta(x) + m\xi(x)\right)$ and $\mathbb{P}(\text{noise} \mid x) = 1 - \mathbb{P}(\text{data} \mid x)$ (Supp. A.1):

$$\theta^* = \arg\min_\theta \left[ -\sum_{i=1}^N \log\left(\frac{q_\theta(s_i)}{q_\theta(s_i) + m\xi(s_i)}\right) - \sum_{i=1}^{mN} \log\left(1 - \frac{q_\theta(t_i)}{q_\theta(t_i) + m\xi(t_i)}\right) \right]. \tag{2}$$

The key advantage of NCE is that the model does not need to be explicitly normalized by the partition function, but nevertheless learns to equal the data distribution $p$ and hence be normalized:

**Theorem 1** (Gutmann & Hyvärinen (2010; 2012)). *Let $\xi$ have full support and suppose there exists some $\theta^*$ such that $q_{\theta^*} = p$. Then $\theta^*$ is a minimum of*

$$\mathcal{L}^{\text{NCE}}(\theta) = -\mathbb{E}_{s \sim p} \log\left(\frac{q_\theta(s)}{q_\theta(s) + m\xi(s)}\right) - m\mathbb{E}_{t \sim \xi} \log\left(1 - \frac{q_\theta(t)}{q_\theta(t) + m\xi(t)}\right) \tag{3}$$

*and the only other extrema of $\mathcal{L}^{\text{NCE}}$ are minima $\tilde\theta$ which also satisfy $q_{\tilde\theta} = p$.*

In NCE, the model typically includes an optimizable normalization parameter $Z$ which we emphasize by writing $q_{\theta,Z} = q_\theta/Z$. But importantly, Thm. 1 applies to any model $q_\theta$ that is able to match the data distribution $p$, even if it does not contain a learnable normalization parameter.

In the setting of learning language models, Jozefowicz et al. (2016) proposed a different version of NCE, called InfoNCE. Instead of classifying samples as data or noise, the aim here is to predict the position of a data sample in an $(m + 1)$-tuple containing $m$ noise samples and one data sample (Supp. A.2). For the uniform noise distribution $\xi$, this yields the expected loss function

$$\mathcal{L}^{\text{InfoNCE}}(\theta) = - \mathop{\mathbb{E}}_{\substack{x \sim p \\ x_1,\ldots,x_m \sim \xi}} \log \left( \frac{q_\theta(x)}{q_\theta(x) + \sum_{i=1}^m q_\theta(x_i)} \right). \tag{4}$$

Ma & Collins (2018) showed that an analogue of Thm. 1 applies to InfoNCE.

## 3.2 Neighbor embeddings

Neighbor embeddings (NE) (Hinton & Roweis, 2002) are a group of dimensionality reduction methods, including UMAP, NCVis, and $t$-SNE that aim to find a low-dimensional embedding $e_1,\ldots,e_n \in \mathbb{R}^d$ of high-dimensional input points $x_1,\ldots,x_n \in \mathbb{R}^D$, with $D \gg d$ and usually $d = 2$ for visualization. NE methods define a notion of similarity over pairs of input points which encodes the neighborhood structure and informs the low-dimensional embedding.

The exact high-dimensional similarity distribution differs between the NE algorithms, but recent work (Böhm et al., 2022; Damrich & Hamprecht, 2021) showed that $t$-SNE and UMAP results stay practically the same when using the binary symmetric $k$-nearest-neighbor graph (s$k$NN) instead of $t$-SNE's Gaussian or UMAP's Laplacian similarities. An edge $ij$ is in s$k$NN if $x_i$ is among the $k$ nearest neighbors of $x_j$ or vice versa. The high-dimensional similarity function is then given by $p(ij) = \mathbb{1}(ij \in \text{s}k\text{NN})/|\text{s}k\text{NN}|$, where $|\text{s}k\text{NN}|$ denotes the number of edges in the s$k$NN graph and $\mathbb{1}$ is the indicator function. NCVis uses the same similarities. We will always use $k = 15$.

There are further differences in the choice of low-dimensional similarity between $t$-SNE and UMAP, but Böhm et al. (2022) showed that they are negligible. Therefore, here we use the Cauchy kernel $\phi(d_{ij}) = 1/(d_{ij}^2 + 1)$ for all NE methods to transform distances $d_{ij} = \|e_i - e_j\|$ in the embedding space into low-dimensional similarities. We abuse notation slightly by also writing $\phi(ij) = \phi(d_{ij})$.

All NE methods in this work can be cast in the framework of parametric density estimation. Here, $p$ is the data distribution to be approximated with a model $q_\theta$, meaning that the space $X$ on which both $p$ and $q_\theta$ live is the set of all pairs $ij$ with $1 \le i < j \le n$. The embedding positions $e_1,\ldots,e_n$ become the learnable parameters $\theta$ of the model $q_\theta$. For $t$-SNE, NC-$t$-SNE (NCVis), and UMAP, $q_\theta$ is proportional to $\phi(\|e_i - e_j\|)$, but the proportionality factor and the loss function are different.

**$t$-SNE** uses MLE and therefore requires a normalized model $q_\theta(ij) = \phi(ij)/Z(\theta)$, where $Z(\theta) = \sum_{k \ne l} \phi(kl)$ is the partition function. The loss function

$$\mathcal{L}^{t\text{-SNE}}(\theta) = -\mathbb{E}_{ij \sim p} \log \left( q_\theta(ij) \right) = - \sum_{i \ne j} \left( p(ij) \log \left( \phi(ij) \right) \right) + \log \left( \sum_{k \ne l} \phi(kl) \right) \tag{5}$$

is the expected negative log-likelihood of the embedding positions $\theta$, making $t$-SNE an instance of MLE. Usually $t$-SNE's loss function is introduced as the Kullback-Leibler divergence between $p$ and $q_\theta$, which is equivalent as the entropy of $p$ does not depend on $\theta$.

**NC-$t$-SNE** uses NCE and optimizes the expected loss function

$$\mathcal{L}^{\text{NC-}t\text{-SNE}}(\theta, Z) = -\mathbb{E}_{ij \sim p} \log \left( \frac{q_{\theta,Z}(ij)}{q_{\theta,Z}(ij) + m\xi(ij)} \right) - m\mathbb{E}_{ij \sim \xi} \log \left( 1 - \frac{q_{\theta,Z}(ij)}{q_{\theta,Z}(ij) + m\xi(ij)} \right), \tag{6}$$

where $q_{\theta,Z}(ij) = \phi(ij)/Z$ with learnable $Z$ and $\xi$ is approximately uniform (see Supp. D).

According to Thm. 1, NC-$t$-SNE has the same optimum as $t$-SNE and can hence be seen as a sampling-based approximation of $t$-SNE. Indeed, we found that $Z$ in NC-$t$-SNE and the partition function $Z(\theta)$ in $t$-SNE converge approximately to the same value (Fig. S10).

**UMAP**'s expected loss function is derived in Damrich & Hamprecht (2021):

$$\mathcal{L}^{\text{UMAP}}(\theta) = -\mathbb{E}_{ij \sim p} \log \left( q_\theta(ij) \right) - m\mathbb{E}_{ij \sim \xi} \log \left( 1 - q_\theta(ij) \right), \tag{7}$$

with $q_\theta(ij) = \phi(ij)$ and $\xi$ is approximately uniform, see Supp. D. This is the effective loss function actually implemented in the UMAP algorithm, but note that it has only about $m/n$ of the repulsion compared to the loss stated in the original UMAP paper (McInnes et al., 2018), as shown in Böhm et al. (2022) and Damrich & Hamprecht (2021) (Supp. C).

In practice, the expectations in UMAP's and NC-$t$-SNE's loss functions are evaluated via sampling, like in Eq. (2). This leads to a fast, $\mathcal{O}(n)$, stochastic gradient descent optimization scheme. Both loss functions are composed of an attractive term pulling similar points (edges of the s$k$NN graph) closer together and a repulsive term pushing random pairs of points further apart. Similarly, $t$-SNE's loss yields attraction along the graph edges while repulsion arises through the normalization term.

## 4 FROM NOISE-CONTRASTIVE ESTIMATION TO NEGATIVE SAMPLING

In this section we work out the precise relationship between NCE and NEG, going beyond prior work (Dyer, 2014; Goldberg & Levy, 2014; Levy & Goldberg, 2014; Ruder, 2016). NEG differs from NCE by its loss function and by the lack of the learnable normalization parameter $Z$. In our setting, NEG's loss function amounts to[1]

$$\mathcal{L}^{\text{NEG}}(\theta) = -\mathbb{E}_{x \sim p} \log\left(\frac{q_\theta(x)}{q_\theta(x) + 1}\right) - m\mathbb{E}_{x \sim \xi} \log\left(1 - \frac{q_\theta(x)}{q_\theta(x) + 1}\right). \tag{8}$$

In order to relate it to NCE's loss function, our key insight is to generalize the latter by allowing to learn a model that is not equal but proportional to the true data distribution.

**Corollary 2.** *Let $\bar{Z}, m \in \mathbb{R}_+$. Let $\xi$ have full support and suppose there exist some $\theta^*$ such that $q_{\theta^*} = \bar{Z}p$. Then $\theta^*$ is a minimum of the generalized NCE loss function*

$$\mathcal{L}_{\bar{Z}}^{\text{NCE}}(\theta) = -\mathbb{E}_{x \sim p} \log\left(\frac{q_\theta(x)}{q_\theta(x) + \bar{Z}m\xi(x)}\right) - m\mathbb{E}_{x \sim \xi} \log\left(1 - \frac{q_\theta(x)}{q_\theta(x) + \bar{Z}m\xi(x)}\right) \tag{9}$$

*and the only other extrema of $\mathcal{L}_{\bar{Z}}^{\text{NCE}}$ are minima $\tilde{\theta}$ which also satisfy $q_{\tilde{\theta}} = \bar{Z}p$.*

*Proof.* The result follows from Thm. 1 applied to the model distribution $\tilde{q}_\theta := q_\theta/\bar{Z}$. □

Dyer (2014) and Ruder (2016) pointed out that for a uniform noise distribution $\xi(x) = 1/|X|$ and as many noise samples as the size of $X$ ($m = |X|$), the loss functions of NCE and NEG coincide, since $m\xi(x) = 1$. However, the main point of NCE and NEG is to use far fewer noise samples in order to attain a speed-up over MLE. Our Cor. 2 for the first time explains NEG's behavior in this more realistic setting ($m \ll |X|$). If the noise distribution is uniform, the generalized NCE loss function with $\bar{Z} = |X|/m$ equals the NEG loss function since $(|X|/m)m\xi(x) = 1$. By Cor. 2, any minimum $\theta^*$ of the NEG loss function yields $q_{\theta^*} = (|X|/m)p$, assuming that there are parameters that make this equation hold. In other words, NEG aims to find a model $q_\theta$ that is proportional to the data distribution with the proportionality factor $|X|/m$ which is typically huge. This is different from NCE, which aims to learn a model equal to the data distribution.

Choosing $m \ll |X|$ does not only offer a computational speed-up but is necessary when optimizing NEG for a neural network with SGD, like we do in Sec.7. Only one single mini-batch is passed through the neural network during each iteration and is thus available for computing the loss. Hence, all noise samples must come from the current mini-batch and their number $m$ is upper-bounded by the mini-batch size $b$. Mini-batches are typically much smaller than $|X|$. Thus, this common training procedure necessitates $m \ll |X|$ highlighting the relevance of Cor. 2.

While NCE uses a model $q_\theta/Z$ with learnable $Z$, we can interpret NEG as using a model $q_\theta/\bar{Z}$ with fixed and very large normalization constant $\bar{Z} = |X|/m$. As a result, $q_\theta$ in NEG needs to attain much larger values to match the large $\bar{Z}$. This can be illustrated in the setting of neighbor embeddings. Applying NEG to the neighbor embedding framework yields an algorithm that we call 'Neg-$t$-SNE'. Recall that in this setting, $|X| = \binom{n}{2}$ is the number of pairs of points. Böhm et al. (2022) found empirically that $t$-SNE's partition function $Z(\theta)$ is typically between $50n$ and $100n$, while in Neg-$t$-SNE, $\bar{Z} = \mathcal{O}(n^2)$ is much larger for modern big datasets. To attain the larger values of $\phi(ij)$ required by NEG, points that are connected in the s$k$NN graph have to move much closer

---

[1] We focus on the loss function, ignoring Mikolov et al. (2013)'s choices specific to word embeddings.

(a) UMAP, no annealing    (b) UMAP with annealing    (c) Neg-$t$-SNE, no ann.    (d) Neg-$t$-SNE with ann.

Figure 2: Embeddings of the MNIST dataset with UMAP and Neg-$t$-SNE with and without learning rate annealing in our implementation. UMAP does not work well without annealing because it implicitly uses the diverging $1/d_{ij}^2$ kernel in NEG, while Neg-$t$-SNE uses the more numerically stable Cauchy kernel (Sec. 6). UMAP's reference implementation also requires annealing, see Figs. S1a, d.

together in the embedding than in $t$-SNE. Indeed, using our PyTorch implementation of Neg-$t$-SNE on the MNIST dataset, we confirmed that Neg-$t$-SNE (Fig. 1d) produced more compact clusters than $t$-SNE (Fig. 1f). See Supp. K and Alg. S1 for implementation details.

We emphasize that our analysis only holds because NEG has no learnable normalization parameter $Z$. If it did, then such a $Z$ could absorb the term $\bar{Z}$ in Eq. (9), leaving the parameters $\theta^*$ unchanged.

## 5   NEGATIVE SAMPLING SPECTRUM

Varying the fixed normalization constant $\bar{Z}$ in Eq. (9) has important practical effects that lead to a whole spectrum of embeddings in the NE setting. The original NEG loss function Eq. (8) corresponds to Eq. (9) with $\bar{Z} = |X|/m$. We still refer to the more general case of using an arbitrary $\bar{Z}$ in Eq. (9) as 'negative sampling' and 'Neg-$t$-SNE' in the context of neighbor embeddings.

Figs. 1a–e show a spectrum of Neg-$t$-SNE visualizations of the MNIST dataset for varying $\bar{Z}$. Per Cor. 2, higher values of $\bar{Z}$ induce higher values for $q_\theta$, meaning that points move closer together. Indeed, the scale of the embedding decreases for higher $\bar{Z}$ as evident from the scale bars in the bottom-right corner of each plot. Moreover, clusters become increasingly compact and then even start to merge. For lower values of $\bar{Z}$ the embedding scale is larger and clusters are more spread out. Eventually, clusters lose almost any separation and start to overlap for very small $\bar{Z}$.

Cor. 2 implies that the partition function $\sum_x q_\theta(x)$ should grow with $\bar{Z}$, and indeed this is what we observed (Fig. 1i). The match between the sum of Cauchy kernels $\sum_{ij} \phi(ij)$ and $\bar{Z}$ was not perfect, but that was expected: The Cauchy kernel is bounded by 1 from above, so values $\bar{Z} > \binom{n}{2}$ are not matchable. Similarly, very small values of $\bar{Z}$ are difficult to match because of the heavy tail of the Cauchy kernel. See Supp. I for a toy example where the match is perfect.

By adjusting the $\bar{Z}$ value, one can obtain Neg-$t$-SNE embeddings very similar to NC-$t$-SNE and $t$-SNE. If the NC-$t$-SNE loss function Eq. (6) has its minimum at some $\theta^*$ and $Z^{\text{NC-}t\text{-SNE}}$, then the Neg-$t$-SNE loss function Eq. (9) with $\bar{Z} = Z^{\text{NC-}t\text{-SNE}}$ is minimal at the same $\theta^*$. We confirmed this experimentally: Setting $\bar{Z} = Z^{\text{NC-}t\text{-SNE}}$, yields a Neg-$t$-SNE embedding (Fig. 1c) closely resembling that of NC-$t$-SNE (NCVis) (Fig. 1g). Similarly, setting $\bar{Z}$ to the partition function $Z(\theta^{t\text{-SNE}})$, obtained by running $t$-SNE, yields a Neg-$t$-SNE embedding closely resembling that of $t$-SNE (Figs. 1b, f). We used $k$NN recall and correlation between pairwise distances for quantitative confirmation (Supp. H).

The Neg-$t$-SNE spectrum is strongly related to the attraction-repulsion spectrum of Böhm et al. (2022). They introduced a prefactor ('exaggeration') to the attractive term in $t$-SNE's loss, which increases the attractive forces, and obtained embeddings similar to our spectrum when varying this parameter. We can explain this as follows. The repulsive term in the NCE loss Eq. (3) has a prefactor $m$ and our spectrum arises from the loss Eq. (9) by varying $\bar{Z}$ in the term $\bar{Z}m$. Equivalently, our spectrum can be obtained by varying the $m$ value (number of noise samples per one s$k$NN edge) while holding $\bar{Z}m$ fixed (Fig. S8). In other words, our spectrum arises from varying the repulsion strength in the contrastive setting, while Böhm et al. (2022) obtained the analogous spectrum by varying the repulsion strength in the $t$-SNE setting (see also Supp. B.2).

## 6 UMAP's CONCEPTUAL RELATION TO $t$-SNE

Our comparison of NEG and NCE in Sec. 4 allows us for the first time to conceptually relate UMAP and $t$-SNE. Originally, McInnes et al. (2018) motivated the UMAP algorithm by a specific choice of weights on the s$k$NN graph and the binary cross-entropy loss. Following LargeVis (Tang et al., 2016), they implemented a sampling-based scheme which they referred to as 'negative sampling', although UMAP's loss function Eq. (7) does not look like the NEG loss Eq. (8). Thus, it has been unclear if UMAP actually uses Mikolov et al. (2013)'s NEG. Here, we settle this question:

**Lemma 3.** *UMAP's loss function (Eq. 7) is NEG (Eq. 8) with the parametric model $\tilde{q}_\theta(ij) = 1/d_{ij}^2$.*

*Proof.* Indeed, $q_\theta(ij) = 1/(1 + d_{ij}^2) = (1/d_{ij}^2)/(1/d_{ij}^2 + 1) = \tilde{q}_\theta(ij)/(\tilde{q}_\theta(ij) + 1)$. □

Lem. 3 tells us that UMAP uses NEG but not with a parametric model given by the Cauchy kernel. Instead it uses a model $\tilde{q}_\theta$, which equals the inverse squared distance between embedding points.

For large embedding distances $d_{ij}$ both models behave alike, but for nearby points they strongly differ: The inverse-square kernel $1/d_{ij}^2$ diverges when $d_{ij} \to 0$, unlike the Cauchy kernel $1/(1+d_{ij}^2)$. Despite this qualitative difference, we found empirically that UMAP embeddings look very similar to Neg-$t$-SNE embeddings at $\bar{Z} = |X|/m$, see Figs. 1d and h for the MNIST example.

To explain this observation, it is instructive to compare the loss terms of Neg-$t$-SNE and UMAP: The attractive term amounts to $-\log\left(1/(d_{ij}^2 + 1)\right) = \log(1 + d_{ij}^2)$ for UMAP and $-\log\left[1/(d_{ij}^2 + 1)/\left(1/(d_{ij}^2 + 1) + 1\right)\right] = \log(2 + d_{ij}^2)$ for Neg-$t$-SNE, while the repulsive term equals $\log\left((1 + d_{ij}^2)/d_{ij}^2\right)$ and $\log\left((2 + d_{ij}^2)/(1 + d_{ij}^2)\right)$, respectively. While the attractive terms are similar, the repulsive term for UMAP diverges at zero but that of Neg-$t$-SNE does not (Fig. S7, Supp. B.2). This divergence introduces numerical instability into the optimization process of UMAP.

We found that UMAP strongly depends on annealing its learning rate down to zero (Figs. 2a, b). Without it, clusters appear fuzzy as noise pairs can experience very strong repulsion and get catapulted out of their cluster (Fig. 2a). While Neg-$t$-SNE also benefits from this annealing scheme (Fig. 2d), it produces a very similar embedding even without any annealing (Fig. 2c). Thus, UMAP's effective choice of the $1/d_{ij}^2$ kernel makes it less numerically stable and more dependent on optimization tricks, compared to Neg-$t$-SNE.[2] See Supp. E for more details.

Our conclusion is that at its heart, UMAP is NEG applied to the $t$-SNE framework. UMAP's sampling-based optimization is much more than a mere optimization trick; it enables us to connect it theoretically to $t$-SNE. When UMAP's loss function is seen as an instance of NEG, UMAP does not use the Cauchy kernel but rather the inverse-square kernel. However, this does not make a strong difference due to the learning rate decay. As discussed in Sec. 4, the fixed normalization constant $\bar{Z}$ in Neg-$t$-SNE / UMAP is much larger than the learned $Z$ in NC-$t$-SNE or $t$-SNE's partition function. This explains why UMAP pulls points closer together than both NC-$t$-SNE and $t$-SNE and is the reason for the typically more compact clusters in UMAP plots (Böhm et al., 2022).

## 7 CONTRASTIVE NE AND CONTRASTIVE SELF-SUPERVISED LEARNING

Contrastive self-supervised representation learning (van den Oord et al., 2018; Tian et al., 2020; He et al., 2020; Chen et al., 2020; Caron et al., 2020) and 'contrastive neighbor embeddings' (a term we suggest for NC-$t$-SNE, Neg-$t$-SNE, UMAP, etc.) are conceptually very similar. The key difference is that the latter use a fixed $k$NN graph to find pairs of similar objects, while the former rely on data augmentations or context to generate pairs of similar objects on the fly. Other differences include the representation dimension ($\sim 128$ vs. 2), the use of a neural network for parametric mapping, the flavor of contrastive loss (InfoNCE vs. NCE / NEG), and the number of noise samples $m$. However, these other differences are not crucial, as we establish here with our unified PyTorch framework.

As an example, we demonstrate that $t$-SNE can also be optimized using the InfoNCE loss, resulting in 'InfoNC-$t$-SNE' (Supp. K and Alg. S1). Its result on MNIST was similar to that of NC-$t$-SNE (Figs. 3b, c). For the default number of noise samples, $m = 5$, the embeddings of both

---

[2]Recently, a pull request to UMAP's GitHub repository effectively changed the kernel of parametric UMAP to the Cauchy kernel, in order to overcome numerical instabilities via an ad hoc fix, see Supp. E.

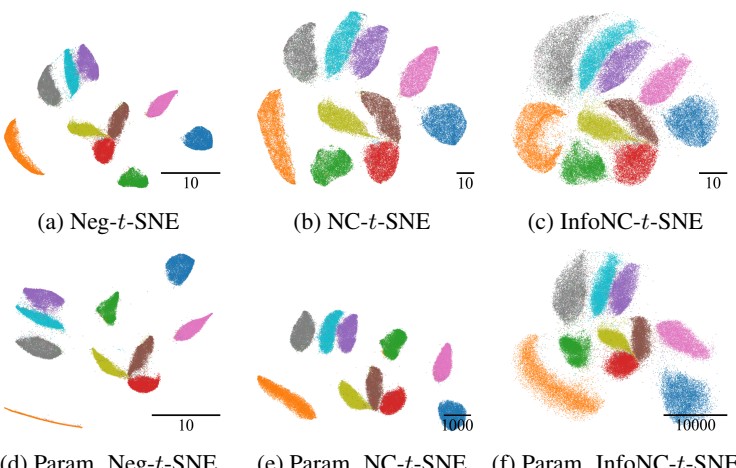

(a) Neg-$t$-SNE      (b) NC-$t$-SNE      (c) InfoNC-$t$-SNE

(d) Param. Neg-$t$-SNE    (e) Param. NC-$t$-SNE    (f) Param. InfoNC-$t$-SNE

Figure 3: NE embeddings of MNIST are qualitatively similar in the non-parametric (top) and parametric settings (bottom). We used our PyTorch framework with $m = 5$ and batch size $b = 1024$.

algorithms were visibly different from $t$-SNE proper (Fig. 1f). Recent work in self-supervised learning (Chen et al., 2020) reports improved performance for more noise samples, in agreement with the theory of Gutmann & Hyvärinen (2012) and Ma & Collins (2018). We observed that both NC-$t$-SNE and InfoNC-$t$-SNE with $m = 500$ approximated $t$-SNE much better (Figs. S18k and S19k). Like $t$-SNE, but unlike the $m = 5$ setting, $m = 500$ required early exaggeration to prevent cluster fragmentation (Figs. S18 and S19). We discuss InfoNC-$t$-SNE's relation to TriMap in Supp. F.

Next, we used our PyTorch framework to obtain parametric versions of all contrastive NE algorithms discussed here (NC-$t$-SNE, InfoNC-$t$-SNE, Neg-$t$-SNE, UMAP). We used a fully connected neural network with three hidden layers as a parametric $\mathbb{R}^D \to \mathbb{R}^d$ mapping and optimized its parameters using Adam (Kingma & Ba, 2015) (Supp. K). We used batch size $b = 1024$ and sampled all $m$ negative samples from within the batch; the dataset was shuffled each epoch before batching. Using all four loss functions, we were able to get parametric embeddings of MNIST that were qualitatively similar to their non-parametric versions (Fig. 3). The parametric versions of NCE and InfoNCE produced much larger embeddings than their non-parametric counterparts, however the final loss values were very similar (Figs. S9a, b). For NCE, the larger scale of the parametric embedding was compensated by a smaller learned normalization parameter $Z$, so that both parametric and non-parametric versions were approximately normalized (Fig. S9c). Our parametric UMAP implementation is very similar to that of Sainburg et al. (2021). But our parametric, approximate $t$-SNE implementations strongly differ from the parametric $t$-SNE of van der Maaten (2009), which constructed separate $k$NN graphs within each batch and optimized the vanilla $t$-SNE loss function.

Neighbor embeddings methods achieve impressive results with only few noise samples, while existent common practice in the self-supervised learning literature (Bachman et al., 2019; Chen et al., 2020; He et al., 2020) recommends very large $m$. As a proof of concept, we demonstrate that using only $m = 16$ non-curated (Wu et al., 2017; Roth et al., 2020) noise samples can suffice for image-based self-supervised learning. We used a SimCLR setup (Chen et al., 2020) to train representations of the CIFAR-10 dataset (Krizhevsky, 2009) using a ResNet18 (He et al., 2016) backbone, a fixed batch size of $b = 1024$ and varying number of noise samples $m$. Other implementation details

Table 1: Test set classification accuracies on CIFAR-10 representations obtained with InfoNCE loss and batch size $b = 1024$ saturate already at a low number of noise samples $m$, common for neighbor embeddings. We used the ResNet18 output $H \in \mathbb{R}^{512}$ and report mean $\pm$ std across three seeds.

|  | $m = 2$ | $m = 16$ | $m = 128$ | $m = 512$ | $m = 2b - 2$ |
|---|---|---|---|---|---|
| $k$NN classifier | $86.9 \pm 0.2$ | $91.7 \pm 0.1$ | $92.0 \pm 0.3$ | $92.2 \pm 0.2$ | $91.9 \pm 0.1$ |
| Linear classifier | $90.7 \pm 0.2$ | $93.1 \pm 0.2$ | $93.3 \pm 0.3$ | $93.2 \pm 0.3$ | $93.3 \pm 0.1$ |

follow Chen et al. (2020)'s CIFAR-10 setup (Supp. K.4). The classification accuracy, measured on the ResNet output, saturates already at $m = 16$ noise samples (Tab. 1). Note that, different from the original SimCLR setup, we decoupled the batch size from the number of noise samples. Recent work found conflicting evidence when varying $m$ for a fixed batch size (Mitrovic et al., 2020; Nozawa & Sato, 2021; Ash et al., 2022). Future research is needed to perform more systematic benchmarks into the importance of $m$. Here, we present these results only as a proof of principle that a similarly low $m$ as in neighbor embeddings (default $m = 5$) can work in a SimCLR setting.

## 8 DISCUSSION AND CONCLUSION

In this work, we studied the relationship between two popular unsupervised learning methods, noise-contrastive estimation (NCE) and negative sampling (NEG). We focused on their application to neighbor embeddings (NE) because this is an active and important application area, but also because NEs allow to directly visualize the NCE / NEG outcome and to form intuitive understanding of how different algorithm choices affect the result. Our study makes three conceptual advances.

First, we showed that NEG replaces NCE's learnable normalization parameter $Z$ by a large constant $\bar{Z}$, forcing NEG to learn a scaled data distribution. While not explored here, this implies that NEG can be used to learn probabilistic models and is not only applicable for embedding learning as previously believed (Dyer, 2014; Ruder, 2016; Le-Khac et al., 2020). In the NE setting, NEG led to the method Neg-$t$-SNE that differs from NCVis / NC-$t$-SNE (Artemenkov & Panov, 2020) by a simple switch from the learnable to a fixed normalization constant. We argued that this can be a useful hyperparameter because it moves the embedding along the attraction-repulsion spectrum similar to Böhm et al. (2022) and hence can either emphasize more discrete structure with higher repulsion (Fig. S16) or more continuous structure with higher attraction (see Figs. S13–S14 for developmental single-cell datasets). Our quantitative evaluation in Supp. H corroborates this interpretation. We believe that inspection of several embeddings along the spectrum can guard against over-interpreting any single embedding. Exploration of the spectrum does not require specialized knowledge. For UMAP, we always have $\bar{Z}^{\text{UMAP}} = \binom{n}{2}/m$; for $t$-SNE, Böhm et al. (2022) found that the partition function $Z^{t\text{-SNE}}$ typically lies in $[50n, 100n]$. Our PyTorch package allows the user to move along the spectrum with a slider parameter s such that s=0 and s=1 correspond to $\bar{Z} = Z^{t\text{-SNE}}$ and $\bar{Z} = \bar{Z}^{\text{UMAP}}$, respectively, without a need for specifying $\bar{Z}$ directly.

A caveat is that Thm. 1 and Cor. 2 both assume the model to be rich enough to perfectly fit the data distribution. This is a strong and unrealistic assumption. In the context of NEs, the data distribution is zero for most pairs of points, which is impossible to match using the Cauchy kernel. Nevertheless, the consistency of MLE and thus $t$-SNE also depends on this assumption. Even without it, the gradients of MLE, NCE, and NEG are very similar (Supp. B). Finally, we validated our Cor. 2 in a toy setting in which its assumptions do hold (Supp. I).

Second, we demonstrated that UMAP, which uses NEG, differs from Neg-$t$-SNE only in UMAP's implicit use of a less numerically stable similarity function. This isolates the key aspect of UMAP's success: Instead of UMAP's appraised (Oskolkov, 2022; Coenen & Pearce, 2022) high-dimensional similarities, the refined Cauchy kernel, or its stated cross-entropy loss function, it is the application of NEG that lets UMAP perform well and makes clusters in UMAP plots more compact and connections between them more continuous than in $t$-SNE, in agreement with Böhm et al. (2022). To the best of our knowledge, this is the first time UMAP's and $t$-SNE's loss functions, motivated very differently, have been conceptually connected.

Third, we argued that contrastive NEs are closely related to contrastive self-supervised learning (SSL) methods such as SimCLR (Chen et al., 2020) which can be seen as parametric InfoNC-$t$-SNE for learning representations in $S^{128}$ based on the unobservable similarity graph implicitly constructed via data augmentations. We feel that this connection has been underappreciated, with the literature on NEs and on self-supervised contrastive learning staying mostly disconnected. Exceptions are the concurrent works of Hu et al. (2023) and Böhm et al. (2023). They propose to use $t$-SNE's Cauchy kernel for SSL. Instead, we bridge the gap between NE and SSL with our InfoNC-$t$-SNE, as well as with parametric versions of all considered NE methods, useful for adding out-of-sample data to an existing visualization. Moreover, we demonstrated that the feasibility of few noise samples in NE translates to the SimCLR setup. We developed a concise PyTorch framework optimizing *all* of the above, which we hope will facilitate future dialogue between the two research communities.

ACKNOWLEDGMENTS

We thank Philipp Berens for comments and support as well as James Melville for discussions.

This research was funded by the Deutsche Forschungsgemeinschaft (DFG, Germany's Research Foundation) via SFB 1129 (240245660; S.D., F.A.H.) as well as via Excellence Clusters 2181/1 "STRUCTURES" (390900948; S.D., F.A.H.) and 2064 "Machine Learning: New Perspectives for Science" (390727645; J.N.B., D.K.), by the German Ministry of Education and Research (Tübingen AI Center, 01IS18039A; J.N.B., D.K.), and by the Cyber Valley Research Fund (D.30.28739; J.N.B.).

The authors thank the International Max Planck Research School for Intelligent Systems (IMPRS-IS) for supporting J.N.B.

ETHICS STATEMENT

Our work analyses connections between popular contrastive learning methods and in particular visualization methods. Visualization methods are fundamental in exploratory data analysis, e.g., in computational biology. Given the basic nature of visualization it can of course potentially be used for malicious goals, but we expect mostly positive societal effects such as more reliable exploration of biological data.

Similarly, contrastive self-supervised learning has the potential to overcome the annotation bottleneck that machine learning faces. This might free countless hours of labelling for more productive use but could also enable institutions with sufficient resources to learn from larger and larger datasets, potentially concentrating power.

REPRODUCIBILITY

All of our results are reproducible. We included proofs for the theoretical statement Cor. 2 and those in Supp. B and D. The datasets used in the experiments including preprocessing steps are described in Supp. J. We discuss the most important implementation details in the main text, e.g., in Sec. 7. An extensive discussion of our implementation is included in Supp. K and in Alg. S1.

Our code and instructions for how to reproduce the experiments are publicly available. We separated the implementation of contrastive neighbor embeddings from the scripts and notebooks needed to reproduce our results, to provide dedicated repositories for people interested in using our method and people wanting to reproduce our results. The contrastive neighbor embedding implementation can be found at `https://github.com/berenslab/contrastive-ne` and the scripts and notebooks for reproducing our results at `https://github.com/hci-unihd/cl-tsne-umap`. The relevant commits in both repositories are tagged `iclr2023`.

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

SUPPLEMENTARY TEXT

## A  PROBABILISTIC FRAMEWORKS OF NCE AND INFONCE

In this section we recap the probabilistic frameworks of NCE (Gutmann & Hyvärinen, 2010; 2012) and InfoNCE (Jozefowicz et al., 2016; van den Oord et al., 2018).

### A.1  NCE

In NCE the unsupervised problem of parametric density estimation is turned into a supervised problem in which the data samples need to be identified in a set $S$ containing $N$ data samples $s_1, \ldots, s_N$ and $m$ times as many noise samples $t_1, \ldots, t_{mN}$ drawn from a noise distribution $\xi$, which can be (but does not have to be) the uniform distribution. In other words, we are interested in the posterior probability $\mathbb{P}(y|x)$ of element $x \in S$ coming from the data ($y = $ data) rather than from the noise distribution ($y = $ noise).

The probability of sampling $x$ from noise, $\mathbb{P}(x|\text{noise})$, is just the noise distribution $\xi$, and similarly $\mathbb{P}(x|\text{data})$ is the data distribution $p$. Since the latter is unknown, it is replaced by the model $q_\theta$. Since $S$ contains $m$ times as many noise samples as data samples, the prior class probabilities are $\mathbb{P}(\text{data}) = 1/(m + 1)$ and $\mathbb{P}(\text{noise}) = m/(m + 1)$. Thus, the unconditional probability of an element of $S$ is $\mathbb{P}(x) = (q_\theta(x) + m\xi(x))/(m + 1)$. The posterior probability for classifying some given element $x$ of $S$ as data rather than noise is thus

$$\mathbb{P}(\text{data}|x) = \frac{\mathbb{P}(x|\text{data})\mathbb{P}(\text{data})}{\mathbb{P}(x)} = \frac{q_\theta(x)}{q_\theta(x) + m\xi(x)}. \tag{10}$$

NCE optimizes the parameters $\theta$ by maximizing the log-likelihood of the posterior class distributions, or, equivalently, by minimizing the negative log-likelihoods. This is the same as a sum over binary cross-entropy losses (Eq. (2) in the main text):

$$\theta^* = \arg\min_\theta \left[ -\sum_{i=1}^{N} \log\left(\frac{q_\theta(s_i)}{q_\theta(s_i) + m\xi(s_i)}\right) - \sum_{i=1}^{mN} \log\left(1 - \frac{q_\theta(t_i)}{q_\theta(t_i) + m\xi(t_i)}\right) \right]. \tag{11}$$

In expectation, we have the loss function (Eq. (4) in the main text)

$$\mathcal{L}^{\text{NCE}}(\theta) = -\mathbb{E}_{s \sim p} \log\left(\frac{q_\theta(s)}{q_\theta(s) + m\xi(s)}\right) - m\mathbb{E}_{t \sim \xi} \log\left(1 - \frac{q_\theta(t)}{q_\theta(t) + m\xi(t)}\right). \tag{12}$$

Since $q_\theta(x)/(q_\theta(x) + m\xi(x)) = 1/\left[1 + \left(q_\theta(x)/(m\xi(x))\right)^{-1}\right]$, NCE's loss function can also be seen as binary logistic regression loss function with $\log\left(\frac{q_\theta(x)}{m\xi(x)}\right)$ as the input to the logistic function:

$$\mathcal{L}^{\text{NCE}}(\theta) = -\mathbb{E}_{s \sim p} \log\left(\sigma\left(\log\left(\frac{q_\theta(s)}{m\xi(s)}\right)\right)\right) - m\mathbb{E}_{t \sim \xi} \log\left(1 - \sigma\left(\log\left(\frac{q_\theta(t)}{m\xi(t)}\right)\right)\right), \tag{13}$$

where $\sigma(x) = 1/(1 + \exp(-x))$ is the logistic function.

### A.2  INFONCE

Again, we are casting the unsupervised problem of density estimation as a supervised classification problem. We consider a tuple of $m + 1$ samples $T = (x_0, \ldots, x_m)$ one of which comes from the data and the rest from the noise distribution. Instead of classifying each sample independently as noise or data (as in Sec. A.1), here we are interested in identifying the position of the single data sample. This allows us to see the problem as a multi-class classification problem with $m + 1$ classes.

Let $Y$ be the random variable that holds the index of the data sample. A priori, we have $\mathbb{P}(Y = k) = 1/(m + 1)$ for all $k = 0, \ldots, m$. Moreover, conditioned on sample $k$ coming from the data distribution, all other samples must come from the noise distribution, i.e., we

have $\mathbb{P}(x_i|Y=k) = \xi(x_i)$ for $i \neq k$. As the data distribution is unknown, we model it with $\mathbb{P}(x_k|Y=k) = q_\theta(x_k)$ as in Sec. A.1. This yields the likelihood of tuple $T$ given the data index $Y=k$

$$\mathbb{P}(T|Y=k) = q_\theta(x_k) \prod_{i \neq k} \xi(x_i) = \frac{q_\theta(x_k)}{\xi(x_k)} \prod_{i=0}^{m} \xi(x_i). \tag{14}$$

Marginalizing over $Y$, we obtain

$$\mathbb{P}(T) = \frac{1}{m+1} \prod_{i=0}^{m} \xi(x_i) \sum_{k=0}^{m} \frac{q_\theta(x_k)}{\xi(x_k)}. \tag{15}$$

Finally, we can compute the posterior via Bayes' rule as

$$\mathbb{P}(Y=k|T) = \frac{\mathbb{P}(T|Y=k)\mathbb{P}(Y=k)}{\mathbb{P}(T)} = \frac{\frac{q_\theta(x_k)}{\xi(x_k)}}{\sum_{i=0}^{m} \frac{q_\theta(x_i)}{\xi(x_i)}} = \frac{q_\theta(x_k)}{\sum_{i=0}^{m} q_\theta(x_i)}, \tag{16}$$

where the last equality only holds for the uniform noise distribution. The InfoNCE loss is the cross-entropy loss with respect to the true position of the data sample, i.e., in expectation and for uniform $\xi$ it reads:

$$\mathcal{L}^{\text{InfoNCE}}(\theta) = - \mathop{\mathbb{E}}_{\substack{x \sim p \\ x_1,\ldots,x_m \sim \xi}} \log \left( \frac{q_\theta(x)}{q_\theta(x) + \sum_{i=1}^{m} q_\theta(x_i)} \right). \tag{17}$$

Similar to how the NCE loss can be seen as binary logistic regression loss function (Sec. A.1), the InfoNCE loss can be viewed as multinomial logistic regression loss function with the terms $\log \left( \frac{q_\theta(x_i)}{\xi(x_i)} \right)$ entering the softmax function.

## B    GRADIENTS

### B.1    GRADIENTS OF MLE, NCE, AND NEG

Here, we compute the gradients of MLE, NCE, and NEG, elaborating the discussion in Artemenkov & Panov (2020). Following the discussion in Secs. 3.2 and 4, we consider a normalized model $q_\theta(x) = \phi_\theta(x)/Z(\theta)$ with partition function $Z(\theta) = \sum_{x'} \phi_\theta(x')$ for MLE, a model $q_{\theta,Z}(x) = \phi_\theta(x)/Z$ with learnable normalization parameter $Z$ for NCE, and a model $q_{\theta,\bar{Z}}(x) = \phi_\theta(x)/\bar{Z}$ with fixed normalization constant $\bar{Z}$ for NEG. We show the results both for general $\bar{Z}$ and noise distribution $\xi$ and for the default value $\bar{Z} = |X|/m$ and a uniform noise distribution $\xi(x) = 1/|X|$.

Thus, the losses are

$$\mathcal{L}^{\text{MLE}}(\theta) = -\mathbb{E}_{x \sim p} \log \left( q_\theta(x) \right), \tag{18}$$

$$\mathcal{L}^{\text{NCE}}(\theta, Z) = -\mathbb{E}_{x \sim p} \log \left( \frac{q_{\theta,Z}(x)}{q_{\theta,Z}(x) + m\xi(x)} \right) - m\mathbb{E}_{x \sim \xi} \log \left( 1 - \frac{q_{\theta,Z}(x)}{q_{\theta,Z}(x) + m\xi(x)} \right), \tag{19}$$

$$\mathcal{L}^{\text{NEG}}(\theta, \bar{Z}) = -\mathbb{E}_{x \sim p} \log \left( \frac{q_{\theta,\bar{Z}}(x)}{q_{\theta,\bar{Z}}(x) + m\xi(x)} \right) - m\mathbb{E}_{x \sim \xi} \log \left( 1 - \frac{q_{\theta,\bar{Z}}(x)}{q_{\theta,\bar{Z}}(x) + m\xi(x)} \right), \tag{20}$$

$$\mathcal{L}^{\text{NEG}}(\theta) = -\mathbb{E}_{x \sim p} \log \left( \frac{\phi_\theta(x)}{\phi_\theta(x) + 1} \right) - m\mathbb{E}_{x \sim \xi} \log \left( 1 - \frac{\phi_\theta(x)}{\phi_\theta(x) + 1} \right). \tag{21}$$

For MLE, we find

$$\frac{d\mathcal{L}^{\text{MLE}}(\theta)}{d\theta} = \frac{d}{d\theta} \left( -\mathbb{E}_{x \sim p} \log \left( q_\theta(x) \right) \right) \tag{22}$$

$$= \frac{d}{d\theta} \left( -\sum_x \left( p(x) \log \left( \phi_\theta(x) \right) \right) + \log \left( \sum_x \phi_\theta(x) \right) \right) \tag{23}$$

$$= -\sum_x \left( \frac{p(x)}{\phi_\theta(x)} \cdot \frac{d\phi_\theta(x)}{d\theta} \right) - \frac{1}{\sum_x \phi_\theta(x)} \cdot \sum_x \frac{d\phi_\theta(x)}{d\theta} \tag{24}$$

$$= -\sum_x \left( \frac{p(x)}{q_\theta(x)} - 1 \right) \cdot \frac{1}{Z(\theta)} \cdot \frac{d\phi_\theta(x)}{d\theta}. \tag{25}$$

For the NCE loss we will use the templates

$$\frac{d\log\left(\frac{f(z)}{f(z)+c}\right)}{dz} = \frac{d\log\left(\frac{1}{1+c/f(z)}\right)}{dz} \tag{26}$$

$$= -\frac{d\log(1+c/f(z))}{dz} \tag{27}$$

$$= -\frac{1}{1+c/f(z)} \cdot \frac{-c}{f(z)^2} \cdot \frac{df(z)}{dz}, \tag{28}$$

$$= \frac{c}{f(z)+c} \cdot \frac{1}{f(z)} \cdot \frac{df(z)}{dz} \tag{29}$$

$$\frac{d\log\left(\frac{c}{f(z)+c}\right)}{dz} = -\frac{d\log\left(f(z)/c+1\right)}{dz} \tag{30}$$

$$= -\frac{1}{f(z)/c+1} \cdot \frac{1}{c} \cdot \frac{df(z)}{dz} \tag{31}$$

$$= -\frac{1}{f(x)+c} \cdot \frac{df(z)}{dz} \tag{32}$$

with some differentiable function $f(z)$ and $c$ independent of $z$.

For the NCE loss we get

$$\frac{d\mathcal{L}^{\text{NCE}}(\theta, Z)}{d\theta} = \frac{d}{d\theta}\left(-\mathbb{E}_{x\sim p}\log\left(\frac{q_{\theta,Z}(x)}{q_{\theta,Z}(x)+m\xi(x)}\right) - m\mathbb{E}_{x\sim\xi}\log\left(1 - \frac{q_{\theta,Z}(x)}{q_{\theta,Z}(x)+m\xi(x)}\right)\right) \tag{33}$$

$$= \frac{d}{d\theta}\left(-\mathbb{E}_{x\sim p}\log\left(\frac{q_{\theta,Z}(x)}{q_{\theta,Z}(x)+m\xi(x)}\right) - m\mathbb{E}_{x\sim\xi}\log\left(\frac{m\xi(x)}{q_{\theta,Z}(x)+m\xi(x)}\right)\right) \tag{34}$$

$$= -\sum_x\left[p(x)\frac{m\xi(x)}{q_{\theta,Z}(x)+m\xi(x)}\frac{1}{q_{\theta,Z}(x)}\frac{dq_{\theta,Z}(x)}{d\theta}\right. \tag{35}$$

$$\left. + m\xi(x)\frac{-1}{q_{\theta,Z}(x)+m\xi(x)}\frac{dq_{\theta,Z}(x)}{d\theta}\right] \tag{36}$$

$$= -\sum_x\left(\frac{p(x)}{q_{\theta,Z}(x)} - 1\right)\frac{m\xi(x)}{q_{\theta,Z}(x)+m\xi(x)}\frac{1}{Z}\frac{d\phi_\theta(x)}{d\theta}. \tag{37}$$

We used our templates at the third equality sign with $f(z) = q_{\theta,Z}(x)$ and $c = m\xi(x)$. The derivation of the gradients for the NEG loss is entirely analogous to that for NCE with $q_{\theta,Z}(x)$ replaced by $q_{\theta,\bar{Z}}(x)$.

Together, the gradients of the three losses are

$$\frac{d\mathcal{L}^{\text{MLE}}(\theta)}{d\theta} = -\sum_{x\in X}\left(\frac{p(x)}{q_\theta(x)} - 1\right) \qquad\qquad \frac{1}{Z(\theta)} \qquad \frac{d\phi_\theta(x)}{d\theta}, \tag{38}$$

$$\frac{d\mathcal{L}^{\text{NCE}}(\theta, Z)}{d\theta} = -\sum_{x\in X}\left(\frac{p(x)}{q_{\theta,Z}(x)} - 1\right) \qquad \frac{m\xi(x)}{q_{\theta,Z}(x)+m\xi(x)} \quad \frac{1}{Z} \qquad \frac{d\phi_\theta(x)}{d\theta}, \tag{39}$$

$$\frac{d\mathcal{L}^{\text{NEG}}(\theta, \bar{Z})}{d\theta} = -\sum_{x\in X}\left(\frac{p(x)}{q_{\theta,\bar{Z}}(x)} - 1\right) \qquad \frac{m\xi(x)}{q_{\theta,\bar{Z}}(x)+m\xi(x)} \quad \frac{1}{\bar{Z}} \qquad \frac{d\phi_\theta(x)}{d\theta}, \tag{40}$$

$$\frac{d\mathcal{L}^{\text{NEG}}(\theta)}{d\theta} = -\sum_{x\in X}\left(\frac{p(x)}{\phi_\theta(x)} - \frac{m}{|X|}\right) \qquad \frac{1}{\phi_\theta(x)+1} \qquad \frac{d\phi_\theta(x)}{d\theta}. \tag{41}$$

The fractions involving $m\xi(x)$ in the gradients of $\mathcal{L}^{\text{NCE}}$ and $\mathcal{L}^{\text{NEG}}$ tend to one as $m \to \infty$ highlighting that a higher number of noise samples improves NCE's approximation of MLE (Gutmann

& Hyvärinen, 2010; 2012; Artemenkov & Panov, 2020). All four gradients are the same up to this factor and the different choice of normalization. In all of them, the models $q_\theta$, $q_{\theta,Z}$, or $q_{\theta,\bar{Z}}$ try to match the data distribution $p$ according to the first factor in each gradient. This similarity of the gradients holds independent of the assumptions of Thm. 1 and Cor. 2. We optimize all losses with stochastic gradient descent and can therefore expect our analysis to hold even if the assumptions are violated.

### B.2 GRADIENTS FOR NEIGHBOR EMBEDDINGS

Here, we rewrite the above gradients for the case of neighbor embeddings. This means that MLE, NCE, and NEG become $t$-SNE, NC-$t$-SNE, and Neg-$t$-SNE, respectively. We include two variants of Neg-$t$-SNE: for a general value of $\bar{Z}$ and for the default value $\bar{Z} = \binom{n}{2}/m$ used by UMAP. In addition, we also show the gradient of UMAP. Recall that for neighbor embedding, we have $X = \{ij|1 \le i < j \le n\}$, $\theta = \{e_1, \ldots, e_n\}$, and $\phi(ij) = 1/(1 + \|e_i - e_j\|^2)$. The noise distributions are close to uniform, see Supp. D, so that $\xi(ij) \approx \binom{n}{2}^{-1}$.

$$\frac{d\mathcal{L}^{t\text{-SNE}}}{de_i} = 2 \sum_{j \ne i} \left( p(ij) - \frac{\phi(ij)}{\sum_{k \ne l} \phi(kl)} \right) \phi(ij)(e_j - e_i) \quad (42)$$

$$\frac{d\mathcal{L}^{\text{NC-}t\text{-SNE}}}{de_i} = 2 \sum_{j \ne i} \underbrace{\frac{m\xi(ij)}{\frac{\phi(ij)}{Z} + m\xi(ij)}}_{\to 1 \text{ for } m \to \infty} \left( p(ij) - \frac{\phi(ij)}{Z} \right) \phi(ij)(e_j - e_i) \quad (43)$$

$$\frac{d\mathcal{L}^{\text{Neg-}t\text{-SNE}}(\bar{Z})}{de_i} = 2 \sum_{j \ne i} \overbrace{\frac{m\xi(ij)}{\frac{\phi(ij)}{\bar{Z}} + m\xi(ij)}} \left( p(ij) - \frac{\phi(ij)}{\bar{Z}} \right) \phi(ij)(e_j - e_i) \quad (44)$$

$$\frac{d\mathcal{L}^{\text{Neg-}t\text{-SNE}}}{de_i} \approx 2 \sum_{j \ne i} \underbrace{\frac{1}{\phi(ij) + 1}}_{\in [0.5, 1]} \left( p(ij) - \frac{\phi(ij)}{\binom{n}{2}/m} \right) \phi(ij)(e_j - e_i) \quad (45)$$

$$\frac{d\mathcal{L}^{\text{UMAP}}}{de_i} = 2 \sum_{j \ne i} \left( p(ij) - \frac{\phi(ij)}{\binom{n}{2}/m} \underbrace{\frac{1}{1 - \phi(ij)}}_{\substack{\text{numerically unstable} \\ \text{for } e_i \approx e_j}} \right) \phi(ij)(e_j - e_i) \quad (46)$$

As above, we see that NC-$t$-SNE and Neg-$t$-SNE have terms that go to one for $m \to \infty$. For Neg-$t$-SNE with default $\bar{Z}$, we used the approximation $\xi(ij) \approx \binom{n}{2}^{-1}$. Here, the corresponding term is restricted to the interval $[0.5, 1]$. More generally, we see that the repulsive force in Neg-$t$-SNE scales as $1/\bar{Z}$. Its default value of $n(n-1)/(2m)$ is much larger than the typical values of the partition function $\sum_{k \ne l} \phi(kl)$ in $t$-SNE or the learned $Z$ of NC-$t$-SNE, which are typically in the range $[50n, 100n]$ (Böhm et al., 2022), leading to much smaller repulsion in default Neg-$t$-SNE and in UMAP than in NC-$t$-SNE or $t$-SNE. Finally, the repulsive part of the UMAP gradient contains a term that diverges for $e_i \approx e_j$, leading to $\phi(ij) \approx 1$, reflecting the discussion of UMAP's numerical instability in Sec. 6.

## C  UMAP'S LOSS FUNCTION

In the original UMAP paper, McInnes et al. (2018) define weights $\mu(ij) \in [0, 1]$ on the s$k$NN graph and state that these shall be reproduced by the low-dimensional similarities $\phi(ij)$ by means of a sum of binary cross-entropy loss functions, one for each edge $ij$

$$-\sum_{ij} \left[ \mu(ij) \log\left(\phi(ij)\right) + \left(1 - \mu(ij)\right) \log\left(1 - \phi(ij)\right) \right]. \quad (47)$$

Indeed, this loss has its minimum at $\mu(ij) = \phi(ij)$ for all $ij$. However, it is of course not possible to achieve zero loss for any real-world data using the Cauchy kernel in two dimensions. Experiments show that using this loss function in practice leads to an excess of repulsion and consequently to very

poor embeddings (Böhm et al., 2022). The actual UMAP implementation has much less repulsion due to the sampling of repulsive edges, see below.

As the weights $\mu(ij)$ are only supported on the sparse $s k$NN graph, most of the $1 - \mu(ij)$ terms are equal to one. To simplify the loss function, the UMAP paper replaces all $1 - \mu(ij)$ terms by $1$, leading to the loss function

$$-\sum_{ij} \Big[ \mu(ij) \log \big( \phi(ij) \big) + \log \big( 1 - \phi(ij) \big) \Big]. \tag{48}$$

In the implementation, UMAP samples the repulsive edges, which drastically changes the loss (Böhm et al., 2022) to the effective loss (Damrich & Hamprecht, 2021)

$$-\sum_{ij} \Big[ \mu(ij) \log \big( \phi(ij) \big) + \frac{m(d_i + d_j)}{2n} \log \big( 1 - \phi(ij) \big) \Big], \tag{49}$$

where $d_i = \sum_j \mu(ij)$ denotes the degree of node $i$ and the number of negative samples $m$ is a hyperparameter. By default, $m = 5$. Since $d_i \approx \log(k)$, the effective loss only has about $m \log(k)/n$ of the repulsion in the originally stated loss function. As a result, the $\mu(ij)$ are not reproduced in the embedding space by this loss function.

We rewrite this effective loss function further to fit into our framework. The attractive prefactors $\mu(ij)$ sum to $\sum_{ij} \mu(ij)$, while the repulsive prefactors add up to $m$ times this factor. Dividing the entire loss function by this term does not change its properties. But then, we can write the prefactors as probability distributions $p(ij) = \mu(ij)/\sum_{ij} \mu(ij)$ and $\xi(ij) = \big( p(i) + p(j) \big)/2n$ using $p(i) = \sum_j p(ij)$. With this, we can write the effective loss function as

$$-\sum_{ij} p(ij) \log \big( \phi(ij) \big) - m \sum_{ij} \xi(ij) \log \big( 1 - \phi(ij) \big), \tag{50}$$

or in the expectation form as

$$-\mathbb{E}_{ij \sim p} \log \big( \phi(ij) \big) - m \mathbb{E}_{ij \sim \xi} \log \big( 1 - \phi(ij) \big), \tag{51}$$

like we do in Eq. (7).

## D    NOISE DISTRIBUTIONS

Here we discuss the various noise distributions used by UMAP, NCVis, and our framework. The main claim is that all these noise distributions are sufficiently close to uniform, even though their exact shape depends on the implementation details.

Since our actual implementation, as well as the reference implementations of UMAP and NCVis, considers edges $ij$ and $ji$ separately, we will do so from now on. Hence, there is now a total of $E := 2|s k$NN$|$ edges. We always assume that $p(ij) = p(ji)$ and adding up the probabilities for both directions yields one: $\sum_{i,j=1}^n p(ij) = 1$. For a given data distribution over pairs of points $ij$, we define $p(i) = \sum_{j=1}^n p(ij)$ so that $\sum_i p(i) = 1$. As discussed in Supp. B of Damrich & Hamprecht (2021), the $p(i)$ values are approximately constant when $p(ij)$ is uniform on the $s k$NN graph or proportional to the UMAP similarities.

### D.1    NOISE DISTRIBUTIONS OF UMAP AND NCVIS

UMAP's noise distribution is derived in Damrich & Hamprecht (2021) and reads in our notation (Supp. C)

$$\xi(ij) = \frac{p(i) + p(j)}{2n}. \tag{52}$$

Note that UMAP uses a weighted version of the $s k$NN graph. Still, $\xi$ is close to uniform, see Supp. B of Damrich & Hamprecht (2021).

The noise distribution of NCVis is also close to being uniform and equals (Artemenkov & Panov, 2020):

$$\xi(ij) = \frac{p(i)}{n} . \tag{53}$$

This is a slightly different noise distribution than in UMAP and in particular it is asymmetric. However, we argue that in practice it is equivalent to the same noise distribution as in UMAP. The noise distribution is used in two ways in NCVis: for sampling noise samples and in the posterior class probabilities

$$\mathbb{P}(\text{data}|ij) = \frac{q_{\theta,Z}(ij)}{q_{\theta,Z}(ij) + m\xi(ij)} . \tag{54}$$

Both in the reference NCVis implementation and in ours, for the second role, the noise distribution is explicitly approximated by the uniform one and we use the posterior probabilities

$$\mathbb{P}(\text{data}|ij) = \frac{q_{\theta,Z}(ij)}{q_{\theta,Z}(ij) + m\frac{1}{2|sk\text{NN}|}} . \tag{55}$$

Together with the symmetry of the Cauchy kernel this implies that the repulsion on the embedding vectors $e_i$ and $e_j$ from noise sample $ij$ and $ji$ is the same. As a result, the expectation

$$\mathbb{E}_{ij \sim \xi} \log \left( 1 - \frac{q_{\theta,Z}(ij)}{q_{\theta,Z}(ij) + m\frac{1}{2|sk\text{NN}|}} \right) \tag{56}$$

is the same for $\xi(ij) = p(i)/n$ and for UMAP's noise distribution

$$\xi(ij) = \frac{p(i) + p(j)}{2n} . \tag{57}$$

### D.2 EFFECT OF BATCHED TRAINING ON THE NOISE DISTRIBUTION

In our framework, the noise distribution is influenced by the batched training procedure (Alg. S1) because the negative samples can come only from the current training batch.

In every epoch, we first shuffle the set of directed edges of the $sk$NN graph and then chunk it into batches. To emphasize, the batches consist of *directed edges* and not of the original data points. For each edge in a batch, we take its head and sample $m$ indices from the heads and tails of all edges in the batch (excluding the already selected head) and use them as tails to form negative sample pairs.

To obtain a negative sample pair $ij$, the batch must contain some directed edge $ik$, providing the head of the negative sample, and some pair $lj$ or $jl$, providing the tail. We want to derive the expected number of times that a directed edge $ij$ is considered as a negative sample in a batch. For simplicity, let us assume that the number of batches divides the number of directed edges $E$. As the set of $sk$NN edges is shuffled every epoch, the expected number of pairs $ij$ as negative samples is the same for all batches.

Let us consider a batch $B$ of size $b$. We denote by $Y_{rs}$ the random variable that holds the number of times edge $rs$ appears in $B$. We also introduce random variables $Y_{\neg rs} = \sum_{t \neq r} Y_{ts}$ and $Y_{r \neg s} = \sum_{t \neq s} Y_{rt}$. Let $p(r \neg s) := p(\neg sr) := p(r) - p(rs)$. For each occurrence of an $i$ as head of an edge in $B$, we sample $m$ tails to create negative samples uniformly from all head and tails in $B$ with replacement, but we prevent sampling the identical head $i$ as negative sample tail. If, however, the same node $i$ is part of the other edges in the batch, then it may be sampled and would create a futile negative sample $ii$. There are $m$ chances for creating a negative sample edge $ij$ for every head $i$ and any occurrence of $j$ in the batch. The number of heads $i$ in the batch is $Y_{ij} + Y_{i \neg j}$ and the number of occurrences of $j$ is $Y_{ij} + Y_{ji} + Y_{\neg ij} + Y_{j \neg i}$. Since we sample the tail of a negative sample pair uniformly with replacement, any of the occurrences of $j$ has probability $1/(2b-1)$ to be selected. Hence, the expected number $N_{ij}$ of times that the ordered pair $ij$ with $i \neq j$ is considered as a negative sample in batch $B$ is

$$N_{ij} = m(Y_{ij} + Y_{i \neg j})\frac{Y_{ij} + Y_{ji} + Y_{\neg ij} + Y_{j \neg i}}{2b - 1} . \tag{58}$$

Since a head $i$ may not choose itself to form a negative sample, the expected number of times that $ii$ appears as negative sample in the batch is

$$N_{ii} = m \sum_j Y_{ij} \frac{Y_{ij} - 1 + Y_{i \neg j} + Y_{ji} + Y_{\neg ji}}{2b - 1}. \tag{59}$$

Since the batches are sampled without replacement, the random variables $Y_{rs}$ are distributed according to a multivariate hypergeometric distribution, so that

$$\mathbb{E}(Y_{rs}) = bp(rs), \tag{60}$$

$$\mathbb{E}(Y_{\neg rs}) = bp(\neg rs), \tag{61}$$

$$\mathrm{Var}(Y_{rs}) = b \frac{E - b}{E - 1} p(rs)\big(1 - p(rs)\big), \tag{62}$$

$$\mathrm{Cov}(Y_{rs}, Y_{\neg uv}) = -b \frac{E - b}{E - 1} p(rs) p(\neg uv). \tag{63}$$

We use these expressions and analogous ones together with the symmetries $p(rs) = p(sr)$ to compute (leaving out intermediate algebra steps) the expectation of $N_{ij}$ over the shuffles:

$$\mathbb{E}(N_{ij}) = \frac{mb}{2b - 1} \left( \frac{E - b}{E - 1} p(ij) + 2 \left( b - \frac{E - b}{E - 1} \right) p(i)p(j) \right). \tag{64}$$

Since we sample $m$ negative samples for each positive sample and since each batch contains $b$ positive samples, we need to divide $\mathbb{E}(N_{ij})$ by $mb$ to obtain $\xi(ij)$:

$$\xi(ij) = \frac{1}{2b - 1} \left( \frac{E - b}{E - 1} p(ij) + 2 \left( b - \frac{E - b}{E - 1} \right) p(i)p(j) \right). \tag{65}$$

Similarly,

$$\mathbb{E}(N_{ii}) = \frac{mb}{2b - 1} \left( -\frac{b - 1}{E - 1} p(i) + 2 \left( b - \frac{E - b}{E - 1} \right) p(i)^2 \right) \tag{66}$$

and hence the noise distribution value for the pair $ii$ is

$$\xi(ii) = \frac{1}{2b - 1} \left( -\frac{b - 1}{E - 1} p(i) + 2 \left( b - \frac{E - b}{E - 1} \right) p(i)^2 \right). \tag{67}$$

We see that the noise distribution depends on the batch size $b$. This is not surprising: For example, if the batch size is equal to one, the ordered pair $ij$ can only be sampled as a negative sample in the single batch that consists of that pair. Indeed, for $b = 1$ our formula yields

$$\xi(ij) = p(ij), \tag{68}$$

meaning that the data and the noise distributions coincide. Conversely, if $b = E$ and there is only one batch, we obtain

$$\xi(ij) = \frac{2E}{2E - 1} p(i)p(j) \tag{69}$$

and the noise distribution is close to uniform. For batch sizes between $1$ and $E$ the noise distribution is in between these two extremes. For MNIST, $E \approx 1.5 \cdot 10^6$, and in our experiments we used $b = 1024$. This means that the prefactor of the share of the data distribution is about $0.0005$ while that of the near-uniform distribution $p(i)p(j)$ is about $0.9995$, so the resulting noise distribution is close to uniform. Note that Thm. 1 and Cor. 2 only require the noise distribution to have full support which is the case for any batch size greater than one.

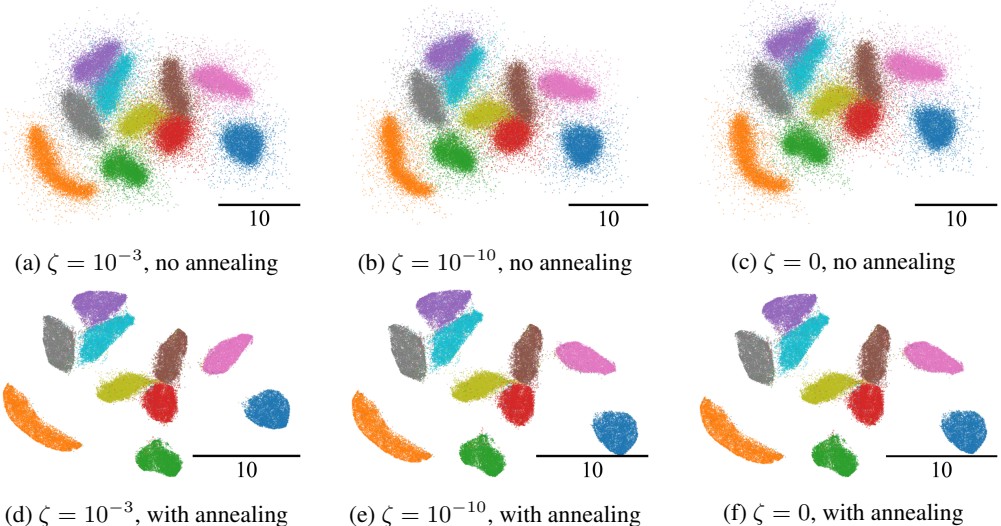

(a) $\zeta = 10^{-3}$, no annealing     (b) $\zeta = 10^{-10}$, no annealing     (c) $\zeta = 0$, no annealing

(d) $\zeta = 10^{-3}$, with annealing     (e) $\zeta = 10^{-10}$, with annealing     (f) $\zeta = 0$, with annealing

Figure S1: UMAP embeddings of the MNIST dataset, ablating numerical optimization tricks of UMAP's reference implementation. The learning rate annealing is crucial (bottom row) but safe-guarding against divisions by zero in UMAP's repulsive term Eq. (71) by adding $\zeta$ to the denominator has little effect. These experiments were run using the reference implementation, modified to change the $\zeta$ value and to optionally switch off the learning rate annealing.

## E OPTIMIZATION TRICKS IN UMAP'S REFERENCE IMPLEMENTATION

UMAP's repulsive term

$$-\log(1 - \phi(ij)) = \log\left(\frac{1 + d_{ij}^2}{d_{ij}^2}\right) \tag{70}$$

can lead to numerical problems if the two points of the negative sample pair are very close. In addition to the learning rate decay, discussed in Sec. 6, UMAP's implementation uses further tricks to prevent unstable or even crashing training.

In non-parametric UMAP's reference implementation, the gradient on embedding position $e_i$ exerted by a single sampled repulsive pair $ij$ is actually

$$2\frac{1}{d_{ij}^2 + \zeta}\frac{1}{1 + d_{ij}^2}(e_j - e_i) \tag{71}$$

with $\zeta = 0.001$ instead of $\zeta = 0$. This corresponds to the full loss function

$$-\mathbb{E}_{ij \sim p} \log(\phi(ij)) - m\left(1 + \frac{\zeta}{1 - \zeta}\right)\mathbb{E}_{ij \sim \xi} \log\left(1 + \frac{\zeta}{1 - \zeta} - \phi(ij)\right). \tag{72}$$

However, we found that $\zeta$ does not have much influence on the appearance of a UMAP embedding. Fig. S1 shows MNIST embeddings obtained using the original UMAP implementation modified to use different values of $\zeta$. Neither a much smaller positive value such as $\zeta = 10^{-10}$ nor setting $\zeta = 0$ substantially changed the appearance of the embedding (even though some runs with $\zeta = 0$ did crash). The learning rate annealing played a much bigger role in how the embedding looked like (Fig. S1, bottom row).

The reference implementation of parametric UMAP uses automatic differentiation instead of implementing the gradients manually. To avoid terms such as $\log(0)$ in the repulsive loss, it clips the argument of the logarithm from below at the value $\varepsilon = 10^{-4}$, effectively using the loss function

$$-\mathbb{E}_{ij \sim p} \log\left(\max\left\{\varepsilon, \frac{1}{1 + d_{ij}^2}\right\}\right) - m\mathbb{E}_{ij \sim \xi} \log\left(\max\left\{\varepsilon, 1 - \frac{1}{1 + d_{ij}^2}\right\}\right). \tag{73}$$

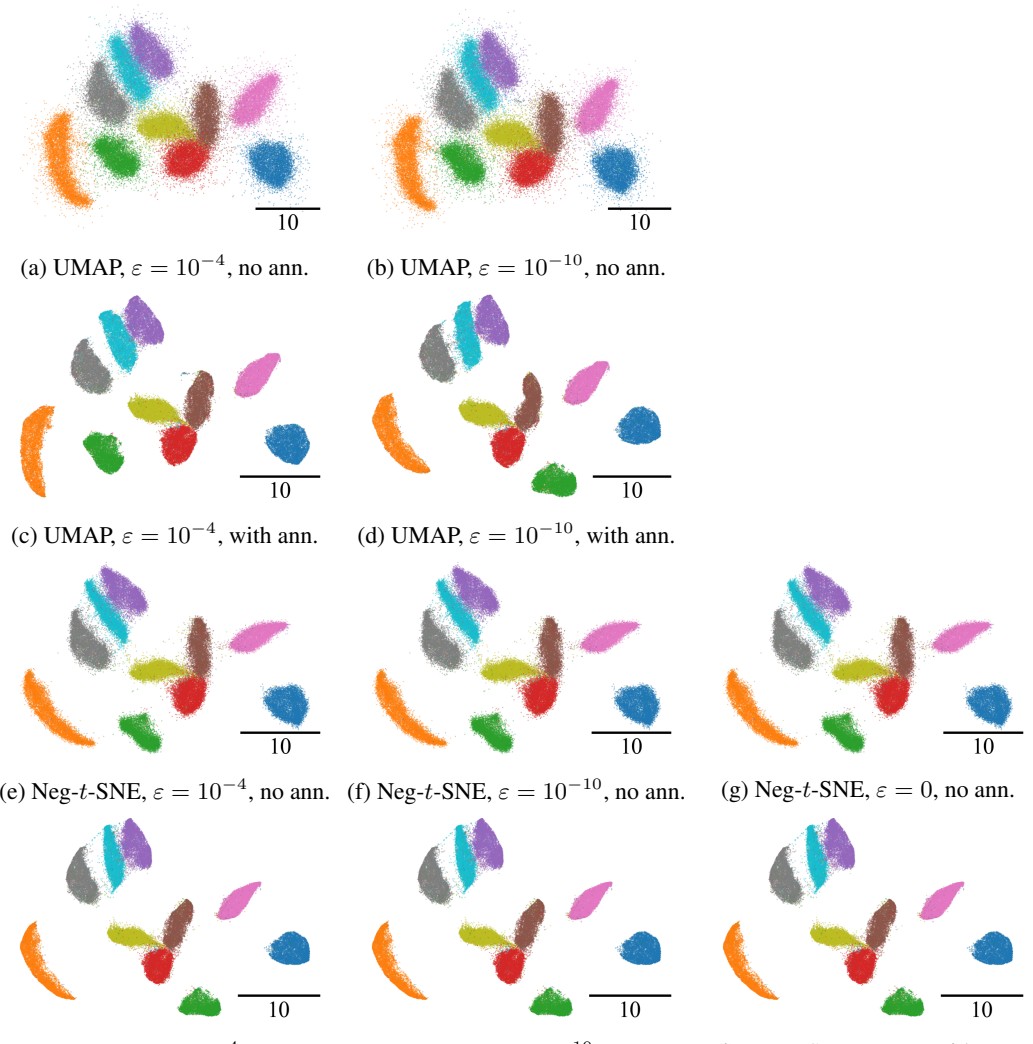

(a) UMAP, $\varepsilon = 10^{-4}$, no ann.    (b) UMAP, $\varepsilon = 10^{-10}$, no ann.

(c) UMAP, $\varepsilon = 10^{-4}$, with ann.    (d) UMAP, $\varepsilon = 10^{-10}$, with ann.

(e) Neg-$t$-SNE, $\varepsilon = 10^{-4}$, no ann.   (f) Neg-$t$-SNE, $\varepsilon = 10^{-10}$, no ann.   (g) Neg-$t$-SNE, $\varepsilon = 0$, no ann.

(h) Neg-$t$-SNE, $\varepsilon = 10^{-4}$, with ann.(i) Neg-$t$-SNE, $\varepsilon = 10^{-10}$, with ann.   (j) Neg-$t$-SNE, $\varepsilon = 0$, with ann.

Figure S2: UMAP and Neg-$t$-SNE embeddings of the MNIST dataset using different values $\varepsilon$ at which we clip arguments to logarithm functions. These experiments were done using our implementation. Varying $\varepsilon$ did not strongly influence the appearance of the embedding. But setting $\varepsilon = 0$ led to crashing UMAP runs. Annealing the learning rate is important for UMAP, yet not for Neg-$t$-SNE.

We employ a similar clipping in our code whenever we apply the logarithm function. Again, we found that the exact value of $\varepsilon$ is not important for our UMAP reimplementation, while using the learning rate annealing is (Fig. S2, top two rows). In the extreme case of setting $\varepsilon = 0$, our UMAP runs crashed. We believe that the reason is that we allow negative sample pairs to be of the form $ii$, which would not send any gradient, but would lead to a zero argument to the logarithm. The reference implementation of UMAP excludes such negative sample pairs $ii$.

Our Neg-$t$-SNE approach does not have any of these problems, as the repulsive term is upper bounded

$$-\log\left(1 - \frac{\frac{1}{1+d_{ij}^2}}{\frac{1}{1+d_{ij}^2}+1}\right) = \log\left(\frac{2+d_{ij}^2}{1+d_{ij}^2}\right) \leq \log(2) \tag{74}$$

and does not diverge for $d_{ij} \to 0$. For this reason, Neg-$t$-SNE is not very sensitive to the value at which we clip arguments to the logarithm and works even with $\varepsilon = 0$, both with and without learning rate annealing (Fig. S2, bottom two rows).

The attractive terms in the loss functions do not pose numerical problems in practice due to the heavy tail of the Cauchy kernel. That said, in order to keep different experiments and losses comparable, we used clipping in both the attractive and the repulsive loss terms with $\varepsilon = 10^{-10}$ for all neighbor embedding plots computed with our framework unless otherwise stated.

A recent pull request (#856) to the parametric part of UMAP's reference implementation proposed another way to ameliorate the numerical instabilities. The clipping of the arguments to the logarithm was replaced with a sigmoid of the logarithm of the Cauchy kernel, so that the attractive and repulsive terms become

$$- \log \Big( \max \big( \varepsilon, \phi(ij) \big) \Big) \to - \log \Big( \sigma \Big( \log \big( \phi(ij) \big) \Big) \Big), \tag{75}$$

$$- \log \Big( \max \big( \varepsilon, 1 - \phi(ij) \big) \Big) \to - \log \Big( 1 - \sigma \Big( \log \big( \phi(ij) \big) \Big) \Big), \tag{76}$$

where $\sigma(x) = 1/\big(1 + \exp(-x)\big)$ is the sigmoid function. This change can seem drastic as, e.g., for $\phi(ij) = 1$ we have $\max \big\{ \varepsilon, \phi(ij) \big\} = 1$, but $\sigma \Big( \log \big( \phi(ij) \big) \Big) = 1/2$. But unravelling the definitions shows that this turns the loss function precisely into our Neg-$t$-SNE loss function since

$$\sigma \Big( \log \big( \phi(ij) \big) \Big) = \frac{1}{1 + \exp \Big( - \log \big( \phi(ij) \big) \Big)} = \frac{1}{1 + \phi(ij)^{-1}} = \frac{\phi(ij)}{\phi(ij) + 1}. \tag{77}$$

So, in order to overcome the numerical problems incurred by UMAP's implicit choice of $1/d_{ij}^2$ as similarity kernel, the pull request suggested a fix that turns out to be equivalent to negative sampling using the Cauchy kernel. We encourage this change to UMAP as it makes its loss function equivalent to our Neg-$t$-SNE and thus also conceptually more related to $t$-SNE. We also suggest to implement it in the non-parametric case.

## F  TriMap and InfoNC-$t$-SNE

The neighbor embedding method TriMap (Amid & Warmuth, 2019) also employs a contrastive technique to compute the layout. TriMap considers triplets $(i, j, k)$, where $i$ and $j$ are data points that are more similar than $i$ and $k$. It moves the embeddings $e_i$ and $e_j$ closer together and pushes $e_i$ and $e_k$ further apart. While a full investigation of TriMap is beyond the scope of our work, here we argue that it bears some similarity to InfoNC-$t$-SNE with $m = 1$.

First, we describe TriMap in more detail. Most of the triplets $(i, j, k)$ consist of nearest neighbors $i$ and $j$ and a random point $k$. By default, 12 nearest neighbors $j$ are chosen for each $i$, using a modified Euclidean distance. For each such pair $(i, j)$ additional distinct points $k$ (by default 4) are chosen randomly from the set of all points excluding $i$ and all the chosen $j$. This leads to 48 triplets $(i, j, k)$ for each point $i$. Additionally, several (by default 3) triplets are formed for each $i$ with random $j$ and $k$ only ensuring that the similarity between $i$ and $j$ is larger than between $i$ and $k$. Thus, for each point $i$ there are in total 51 triplets $(i, j, k)$ leading to a grand total of $51n$ triplets. These triplets are formed ahead of the layout optimization phase and kept fixed throughout. TriMap also assigns a weight $w_{ijk}$ to each triplet. This weight is larger for triplets in which $i$ and $j$ are much more similar to each other than $i$ and $k$.

In our notation, TriMap's loss function can be written as

$$\sum_{ijk \in \text{triplets}} w_{ijk} \frac{\phi(ik)}{\phi(ij) + \phi(ik)} = \text{const} - \sum_{ijk \in \text{triplets}} w_{ijk} \frac{\phi(ij)}{\phi(ij) + \phi(ik)}. \tag{78}$$

This loss is very similar to the loss of InfoNC-$t$-SNE with $m = 1$:

$$- \mathop{\mathbb{E}}_{\substack{ij \sim p \\ k \sim \mathcal{U}}} \log \left( \frac{\phi(ij)}{\phi(ij) + \phi(ik)} \right). \tag{79}$$

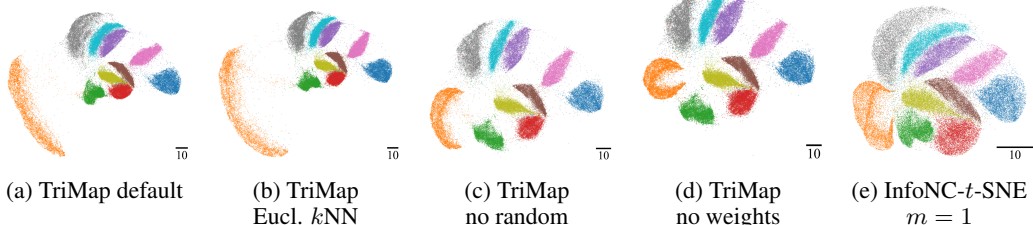

|     |     |     |     |     |
| :-: | :-: | :-: | :-: | :-: |
| (a) TriMap default | (b) TriMap Eucl. $k$NN | (c) TriMap no random | (d) TriMap no weights | (e) InfoNC-$t$-SNE $m = 1$ |

Figure S3: Ablating design choices of TriMap on the MNIST dataset. Simplifying TriMap leads to an embedding similar to that of InfoNC-$t$-SNE with $m = 1$. The biggest difference results from the omission of random triplets in panel c.

Table S1: Quality of TriMap and InfoNC-$t$-SNE. Means and standard deviations over three runs.

|                              | $k$NN recall      | Spearman correlation |
| ---------------------------- | ----------------- | -------------------- |
| TriMap (default)             | $0.098 \pm 0$     | $0.245 \pm 0.005$    |
| TriMap ($k$NN)               | $0.099 \pm 0$     | $0.232 \pm 0.003$    |
| TriMap (no random)           | $0.101 \pm 0$     | $0.355 \pm 0.004$    |
| TriMap (no weights)          | $0.099 \pm 0$     | $0.381 \pm 0.004$    |
| InfoNC-$t$-SNE ($m = 1$)     | $0.085 \pm 0.001$ | $0.381 \pm 0.006$    |
| InfoNC-$t$-SNE ($m = 5$)     | $0.104 \pm 0$     | $0.365 \pm 0.005$    |

The differences are: the absence of $\log$; the presence of weights; the presence of random triplets; nearest neighbors being based not exactly on Euclidean metric; and fixed triplets during optimization. In the following we modify the reference TriMap implementation and ablate these differences in order to see which ones are more or less important (Figure S3). We measure the quality of all embeddings using two metrics (Tab. S1): as a global metric we use the Spearman correlation between high- and low-dimensional pairwise distances between $5000$ randomly chosen points (Kobak & Berens, 2019) and as a local metric we use $k$NN recall with $k = 15$ (Kobak & Berens, 2019).

First, we modified TriMap to use the $15$ nearest neighbors in Euclidean metric. This did not change the visual appearance of the MNIST embedding (Figs. S3a, b). Next, we additionally omitted the random triplets. The resulting embedding became visually closer to the InfoNC-$t$-SNE embedding (Fig. S3c). Finally, additionally setting all the weights $w_{ijk}$ to one had only a small visual effect (Fig. S3d). Together, these simplifications noticeably improved the global metric compared to the default TriMap, while leaving the local metric almost unchanged (Tab. S1).

Similar to the experiments of Wang et al. (2021), our ablations made TriMap look very similar to InfoNC-$t$-SNE with $m = 1$ (Fig. S3e), suggesting that the remaining differences between them do not play a major role. This refers to the details of the optimization (e.g. fixed triplets vs. randomly sampled triplets in each epoch, full-batch vs. stochastic gradient descent, momentum and adaptive learning rates, etc.), as well as to the presence/absence of the logarithm in the loss function. Since

$$\frac{d \left( \log \left( \frac{\phi(ij)}{\phi(ij) + \phi(ik)} \right) \right)}{d\theta} = \frac{\phi(ij) + \phi(ik)}{\phi(ij)} \frac{d \left( \frac{\phi(ij)}{\phi(ij) + \phi(ik)} \right)}{d\theta}, \tag{80}$$

the gradients of TriMap and InfoNC-$t$-SNE point in the same direction, but are differently scaled. InfoNC-$t$-SNE prioritizes triplets with high $(\phi(ij) + \phi(ik))/\phi(ij)$, i.e. triplets in which the embedding similarity does not respect the triplet well. Note that this is conceptually different from TriMap's weights $w_{ijk}$ which are based on the high-dimensional similarities. However, at least on MNIST, this difference in gradients did not have a major effect (Figs. S3d, e).

Note that our default for InfoNC-$t$-SNE is $m = 5$ instead of $m = 1$. On MNIST, this setting yields a better local score and a similar global score compared to $m = 1$, and much better scores than the default TriMap (Tab. S1). The work of Ma & Collins (2018) theoretically underpins our observation in Fig. S19 that the higher $m$ gets, the better InfoNC-$t$-SNE approximates $t$-SNE.

## G    RELATION TO OTHER VISUALIZATION METHODS

### G.1    $t$-SNE APPROXIMATIONS

A vanilla implementation of $t$-SNE has to compute the partition function $Z(\theta)$ and hence scales quadratically with the sample size $n$. Therefore, nearly all modern $t$-SNE packages implement approximations. Common are Barnes-Hut-$t$-SNE (BH-$t$-SNE) (Yang et al., 2013; van der Maaten, 2014) and FFT-accelerated interpolation-based $t$-SNE (FI$t$-SNE) (Linderman et al., 2019). Both methods approximate the repulsive interaction between all pairs of points and the computation of the partition function, achieving complexities $\mathcal{O}(kn\log(n)2^d)$ and $\mathcal{O}(kn2^d)$, respectively. In fact, all our $t$-SNE plots are done with the FI$t$-SNE implementation of openTSNE (Poličar et al., 2019).

The focus of our work is not primarily on accelerating $t$-SNE, but on understanding its relation to contrastive neighbor embeddings. Different from the $t$-SNE approximations above, these do not approximately compute the partition function, but either learn it (NC-$t$-SNE), use a fixed normalization parameter (Neg-$t$-SNE and UMAP), or do not require it at all (InfoNC-$t$-SNE). They only sample $nm$ repulsive interactions per epoch. This cuts their complexity down to $\mathcal{O}(kmnd)$. In particular, they scale linearly with the embedding dimension $d$ and can therefore be used for more general representation learning with $d \gg 2$, unlike BH-$t$-SNE and FI$t$-SNE.

### G.2    OTHER CONTRASTIVE $t$-SNE APPROXIMATIONS

In addition to NC-$t$-SNE and InfoNC-$t$-SNE, there are two other, recent sampling-based strategies to approximate $t$-SNE, GDR-$t$-SNE (Draganov et al., 2022) and SCE (Yang et al., 2023). Both sample $m$ repulsive interactions per each attractive interaction. In order to scale these repulsive interactions properly, an estimate of the partition function is computed.

In GDR-$t$-SNE, the sum of the Cauchy kernels of the $mkn$ repulsive interactions over each epoch are added up to obtain an estimate of $Z^{t\text{-SNE}}(\theta) \cdot mkn/\big(n(n-1)\big)$. Since both the number of repulsive interactions and the estimate of the partition function are decreased by a factor of $mk/(n-1)$ compared to the vanilla $t$-SNE, the overall repulsion strength is correct. In contrast, SCE (Yang et al., 2023) initializes its estimate of the partition function with the maximum value $n(n-1)$ and then updates it using a moving average after each sampled repulsive pair. Both Draganov et al. (2022) and Yang et al. (2023) use $m = 1$. Their approaches are similar to NCE in that they approximately estimate $Z$ using the sampled noise points, while NCE treats $Z$ as learnable parameter and performs gradient descent on it. Note, however, that the second factor of the NCE gradient in Eq. (39) is not present in the approaches of Draganov et al. (2022) and Yang et al. (2023).

### G.3    LARGEVIS

LargeVis (Tang et al., 2016) was, to the best of our knowledge, the first contrastive neighbor embedding method. It uses NEG. Since it was quickly overshadowed by very similar UMAP, we focused the exposition in the main paper on UMAP. Damrich & Hamprecht (2021) computed LargeVis' effective loss function in closed form. In our notation it reads

$$\mathcal{L}^{\text{LargeVis}}(\theta) = -\mathbb{E}_{ij\sim p}\log(\phi(ij)) - \gamma m \mathbb{E}_{ij\sim\xi}\log(1-\phi(ij)). \qquad (81)$$

The are some implementation differences between LargeVis and UMAP, but in terms of the loss function, the main difference is the $\gamma$ factor in front of the repulsive term, which defaults to 7. Therefore, our analysis of UMAP carries over to LargeVis: Its loss is essentially an instance of NEG with inverse square kernel. The additional factor $\gamma = 7$ moves it along the attraction-repulsion spectrum towards more repulsion, that is, towards $t$-SNE.

### G.4    PACMAP

Pairwise Controlled Manifold Approximation Projection (PaCMAP) (Wang et al., 2021) is a recent visualization method also based on sampling $m$ repulsive forces per one attractive force (with $m = 2$ by default). It employs a large number of additional design choices, such as weak attraction between 'mid-near' points in addition to attraction between nearest neighbors (which are based on a modified Euclidean distance, as in TriMap), several optimization regimes with dynamically changing loss weights, etc. Nevertheless, the core parts of its loss can be related to (generalized) Neg-$t$-SNE.

The attractive loss in PaCMAP is

$$\frac{d^2 + 1}{d^2 + 1 + c} = 1 - c \cdot \frac{1}{d^2 + 1 + c} \tag{82}$$

with $c = 10$ for nearest neighbors (and $c = 10\,000$ for mid-near points but that part of the loss is switched off during the final stage of the optimization). The attractive loss for Neg-$t$-SNE with uniform noise distribution and $\bar{Z} = c|X|/m$ is

$$-\log\left(\frac{\phi(d)}{\phi(d) + c}\right) = -\log\left(\frac{1}{(d/\sqrt{c})^2 + 1 + c}\right). \tag{83}$$

Thus, the differences are the logarithm and some rescaling of the distances. Note that Neg-$t$-SNE with $c = 10$ typically produces very similar result compared to the default $c = 1$ (cf. Fig 1).

The repulsive loss in PaCMAP is

$$\frac{1}{2 + d^2} = 1 - \frac{1 + d^2}{2 + d^2}. \tag{84}$$

Again it is similar to the repulsive loss of Neg-$t$-SNE, but now for $\bar{Z} = |X|/m$, the default negative sampling value also used by UMAP:

$$-\log\left(1 - \frac{\phi(d)}{\phi(d) + 1}\right) = -\log\left(\frac{1}{\phi(d) + 1}\right) = -\log\left(\frac{1 + d^2}{2 + d^2}\right). \tag{85}$$

Here, the difference is just in the logarithm, similar to the connection between TriMap and InfoNC-$t$-SNE in Supp. F.

Empirically, PaCMAP embeddings of high-dimensional data like MNIST look similar to UMAP embeddings, and hence to Neg-$t$-SNE embeddings, in agreement with our analysis.

## H  QUANTITATIVE EVALUATION

This section provides a quantitative evaluation of the Neg-$t$-SNE spectrum. For each embedding on the spectrum, we compute two metrics. The first metric, $k$NN recall with $k = 15$ (Kobak & Berens, 2019), measures the fraction of nearest neighbors in the embedding that are also nearest neighbors in the reference configuration. It is therefore a measure of local quality. The second metric is the Spearman correlation between the pairwise distances in the embedding and in the reference configuration (Kobak & Berens, 2019). To speed up the computation, we consider all pairwise distances between a sample of 5000 points. Since most random pairs of points are not close to each other, this is a measure of global quality.

As reference configuration, we use three different layouts for each dataset. We compare against the original high-dimensional data to measure how faithful Neg-$t$-SNE embeddings are. Additionally, we compare to the corresponding $t$-SNE and UMAP embeddings to investigate for which $\bar{Z}$ value Neg-$t$-SNE matches $t$-SNE and UMAP the best.

We found that compared to the high-dimensional data, the $k$NN recall followed an inverse U-shape across the spectrum (Fig. S4a). $k$NN recall was low for very large and for very small values of $\bar{Z}$. It peaked when $\bar{Z}$ was close to $t$-SNE's partition function $Z^{t\text{-SNE}}$ and NC-$t$-SNE's learned normalization parameter $Z^{\text{NC-}t\text{-SNE}}$. At UMAP's normalization constant $\bar{Z}^{\text{UMAP}}$ the $k$NN recall was usually lower. This confirms our observation that the local quality of the embedding improves when decreasing $\bar{Z}$ from $\bar{Z}^{\text{UMAP}}$ to $Z^{t\text{-SNE}}$ in agreement with Böhm et al. (2022). The $k$NN recall for the Kuzushiji-49 dataset at $\bar{Z} = Z^{t\text{-SNE}}$ was very low, compared to other datasets. We suspect that this was due to incomplete convergence (Fig. S17, Tab. S4).

Conversely, the Spearman correlation mostly increased with $\bar{Z}$ (Fig. S4b), which aligns with our finding that higher attraction improves the global layout of the embedding.

We also computed the $k$NN recall and the Spearman correlation for the proper $t$-SNE and UMAP embeddings (not depicted in Figs. S4a, b). The $k$NN recall was higher for $t$-SNE embeddings than for Neg-$t$-SNE at $\bar{Z} = Z^{t\text{-SNE}}$ (see, e.g., Fig. S18l), likely because proper $t$-SNE considers the repulsive interaction between all points. The metrics for the UMAP embeddings and the Spearman

correlation for $t$-SNE were close to the corresponding values for Neg-$t$-SNE at the respective $\bar{Z}$ values.

When comparing Neg-$t$-SNE embeddings to the proper $t$-SNE embedding, the best fit in terms of $k$NN recall (Fig. S4c) and in terms of Spearman correlation (Fig. S4d) was usually achieved when $\bar{Z}$ was close to $Z^{t\text{-SNE}}$, in accordance with our theory. Again, the Kuzushiji-49 dataset was an outlier, likely due to convergence issues.

Finally, when comparing Neg-$t$-SNE embeddings to the UMAP embedding, the highest Spearman correlation was achieved by $\bar{Z} \approx \bar{Z}^{\text{UMAP}}$ (Fig. S4f), in agreement with our theoretical predictions. The highest $k$NN recall was typically achieved at a slightly lower $\bar{Z}$ (Fig. S4e).

Overall, these experiments provide empirical support for our interpretation of the Neg-$t$-SNE spectrum as implementing a trade-off between global and local structure preservation, and they confirm the predicted locations of $t$-SNE- and UMAP-like embeddings on the spectrum.

We computed the $k$NN recall and the Spearman correlation also for the embeddings in Figs. S18 and S19, where we varied the number of noise samples $m$ and the initialization method for NC-$t$-SNE and InfoNC-$t$-SNE. As expected, we found that higher $m$ improves the $k$NN recall, but $t$-SNE proper achieved higher $k$NN recall still. The Spearman correlation decreased with $m$ when we initialized randomly and did not change much for other initialization methods. Random initialization hurt the Spearman correlation significantly.

# I   Toy dataset

Cor. 2 predicts that the partition function of a Neg-$t$-SNE embedding should equal the $\bar{Z}$ value. In our real-world examples (panels i in Figs. 1, S11–S16) we observed a monotone relationship between them, but not a perfect agreement. As discussed in Sec. 5, this is due to the shape of the Cauchy kernel and the fact that the data distribution is zero for many pairs of points. Here, we consider a toy example for which we observe a perfect match between Neg-$t$-SNE's partition function and the $\bar{Z}$ value, confirming our Cor. 2.

As we operate with the binary s$k$NN graph, the only possible high-dimensional similarities are zero and one. Due to the heavy tail of the Cauchy kernel, we would like our toy example to have no pairs of points for which the data distribution is zero. Thus, all pairs of points need to have equal high-dimensional similarity. In two dimensions, only up to three points can be placed equidistantly from each other. Therefore, we consider the simple case of placing three points according to the Neg-$t$-SNE loss function with $p \equiv \frac{1}{6}$ (there are six directed edges) for various values of $\bar{Z}$. We keep the Cauchy kernel as similarity function, which has maximum $1/(1 + 0^2) = 1$. Thus, it is not possible to match values above $\bar{Z} = 6$, and the three points end up having the same position in the embedding. But for smaller values of $\bar{Z}$, the resulting partition function is indeed exactly equal to $\bar{Z}$ (Fig. S5).

For this experiment we decreased the batch size to 6 and the learning rate to $0.01$, but kept all other hyperparameters at their default values.

# J   Datasets

We used the well-known MNIST (LeCun et al., 1998) dataset for most of our experiments. We downloaded it via the torchvision API from `http://yann.lecun.com/exdb/mnist/`. This website does not give a license. But `https://keras.io/api/datasets/mnist/` and `http://www.pymvpa.org/datadb/mnist.html` name Yann LeCun and Corinna Cortes as copyright holders and claim MNIST to be licensed under CC BY-SA 3.0, which permits use and adaptation. The MNIST dataset consists of $70\,000$ grayscale images, $28 \times 28$ pixels each, that show handwritten digits.

The Kuzushiji-49 dataset (Tarin et al., 2018) was downloaded from `https://github.com/rois-codh/kmnist` where it is licensed under CC-BY-4.0. It contains $270\,912$ grayscale images, $28 \times 28$ pixels each, that show $49$ different cursive Japanese characters.

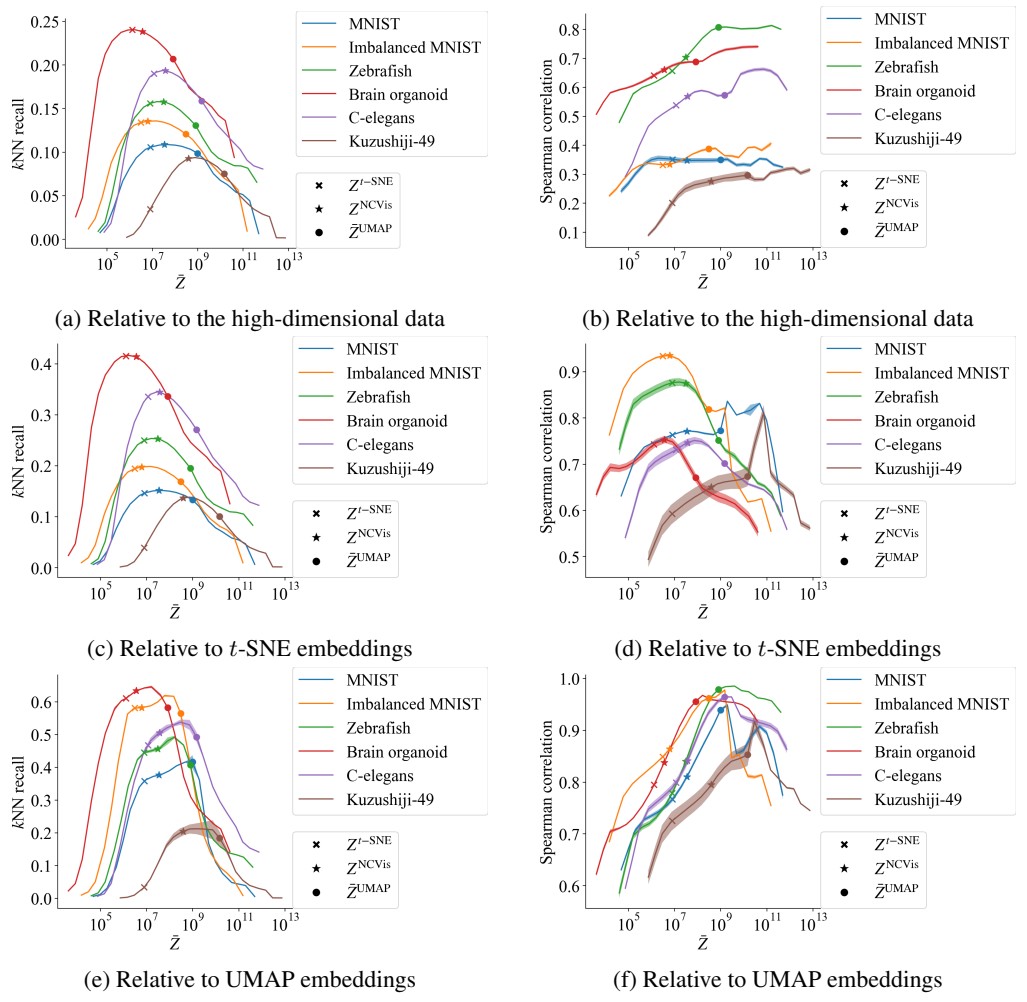

Figure S4: Quantitative evaluation of embeddings on the Neg-$t$-SNE spectrum with respect to different reference configurations. Means and standard deviations over three random seeds are plotted. **Left column:** $k$NN recall, a measure of local agreement. **Right column:** Spearman correlation, a measure of global agreement. **First row:** Comparison of Neg-$t$-SNE embeddings to the high-dimensional input data. **Second row:** Comparison of Neg-$t$-SNE embeddings to the corresponding $t$-SNE embedding. **Third row:** Comparison of Neg-$t$-SNE embeddings to the corresponding UMAP embedding.

The SimCLR experiments were performed on the CIFAR-10 (Krizhevsky, 2009) dataset, another standard machine learning resource. We downloaded it via the `sklearn.datasets.fetch_openml` API from `https://www.openml.org/search?type=data&sort=runs&id=40927&status=active`. Unfortunately, we were not able to find a license for this dataset. CIFAR-10 consists of $60\,000$ images, $32 \times 32$ RGB pixels each, depicting objects from five animal and five vehicle classes.

The transcriptomic dataset of Kanton et al. (2019) was downloaded from `https://www.ebi.ac.uk/arrayexpress/experiments/E-MTAB-7552/`, which permits free use. We only used the $20\,272$ cells in the human brain organoid cell line '409b2'. The transcriptomic dataset of Wagner et al. (2018) was downloaded in its scanpy version from `https://kleintools.hms.harvard.edu/paper_websites/wagner_zebrafish_timecourse2018/mainpage.html`. The full dataset at `https://www.ncbi.nlm.nih.gov/geo/query/acc.cgi?acc=GSE112294` is free to download and reproduce. The dataset contains gene expressions for $63\,530$ cells from a developing zebrafish embryo. The downloaded UMIs of both datasets were preprocessed as in (Böhm et al., 2022; Kobak

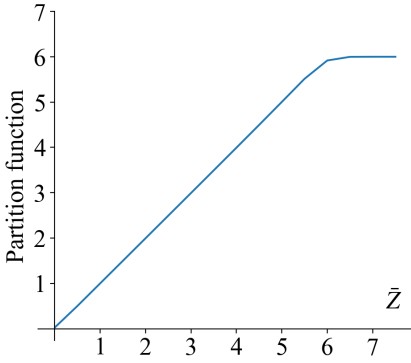

Figure S5: Partition function of Neg-$t$-SNE embeddings as a function of $\bar{Z}$ for a toy dataset with three equidistant points. There is a perfect fit between $\bar{Z}$ and the partition function until $\bar{Z} = 6$, which is the largest partition function a set of three points with six directed edges can have under the Cauchy kernel. Beyond this value, all three points overlap in the embedding. Mean with standard deviation over three random seeds is plotted.

& Berens, 2019). After selecting the 1000 most variable genes, we normalized the library sizes to the median library size in the dataset, log-transformed the normalized values with $\log_2(x + 1)$, and finally reduced the dimensionality to 50 via PCA.

The transcriptomic dataset of the C. elegans flatworm (Packer et al., 2019; Narayan et al., 2021) was obtained from `http://cb.csail.mit.edu/cb/densvis/datasets/` with consent of the authors who license it under CC BY-NC 2.0. It is already preprocessed to 100 principal components.

## K IMPLEMENTATION

### K.1 PACKAGES

All contrastive embeddings were computed with our PyTorch (Paszke et al., 2019) implementation of Neg-$t$-SNE, NC-$t$-SNE, UMAP, and InfoNC-$t$-SNE. Exceptions are Fig. 1 and analogous Figs. S11 – S16. There, for panel g we used the reference implementation of NCVis (Artemenkov & Panov, 2020) (with a fixed number of noise samples $m$, and not the default schedule), and for panel h we used UMAP 0.5. The $t$-SNE plots were created with the openTSNE (Poličar et al., 2019) (version 0.6.1) package. Similarly, we used the reference UMAP implementation in Fig. S1 and openTSNE in Figs. S18 and S19. The TriMap plots in Supp. F were computed with version 1.1.4 of the TriMap package by Amid & Warmuth (2019).

We extended these implementations of NCVis, UMAP, and $t$-SNE to make them accept custom embedding initializations and unweighted $skNN$ graphs and to log various quantities of interest. We always used the standard Cauchy kernel for better comparability.

### K.2 INITIALIZATION, EXAGGERATION AND $skNN$ GRAPH

All PCAs were computed with sklearn (Pedregosa et al., 2011). We used PyKeOps (Charlier et al., 2021) to compute the exact $skNN$ graph and to handle the quadratic complexity of computing the partition functions on a GPU. The same PCA initialization and $skNN$ graph with $k = 15$ were used for all embeddings. The $skNN$ graph for MNIST and Kuzushiji-49 was computed on a 50-dimensional PCA of the dataset.

We initialized all embeddings with a scaled version of PCA (save for Figs. S18 and S19). For $t$-SNE embeddings we rescaled the initialization so that the first dimension has a standard deviation of 0.0001 (as is default in openTSNE), for all other embeddings to a standard deviation of 1. For TriMap, we stuck to its default of scaling the PCA down by a factor of 0.01.

We employed some version of 'early exaggeration' (van der Maaten & Hinton, 2008) for the first $250$ epochs in most non-parametric plots. For $t$-SNE it is the default early exaggeration of openTSNE. When varying $\bar{Z}$ in non-parametric Neg-$t$-SNE, early exaggeration meant using $\bar{Z} = |X|/m$ for the first $250$ epochs (save for Fig. S11). When varying the number of noise samples in Figs. S8, S18 and S19, we still used $m = 5$ for the first $250$ epochs. In Figs. 2, 3, S1, and S2 as well as in all reference NCVis or UMAP plots, we did not use early exaggeration as neither small $\bar{Z}$ nor high $m$ made it necessary. When we used some form of early exaggeration and learning rate annealing, the annealing to zero took place over the first $250$ epochs, was then reset, and annealed again to zero for the remaining, typically $500$, epochs.

### K.3  OTHER DETAILS FOR NEIGHBOR EMBEDDING EXPERIMENTS

When computing logarithms during the optimization of neighbor embeddings, we clip the arguments to the range $[10^{-10}, 1]$, save for Fig. S2, where we varied this lower bound. The lower bound is smaller than in the reference implementation of parametric UMAP, where it is set to $10^{-4}$.

Our defaults were a batch size of $1024$, linear learning rate annealing from $1$ (non-parametric) or $0.001$ (parametric) to $0$ (save for Figs. 2, S1, and S2), $750$ epochs (save for Figs. S10 and S17) and $m = 5$ noise samples (save for Figs. S8, S10, S18, and S19).

Non-parametric runs were optimized with SGD without momentum and parametric runs with the Adam optimizer (Kingma & Ba, 2015). Parametric runs used the same feed-forward neural net architecture as the reference parametric UMAP implementation. That is, four layers with dimensions *input dimension* $- 100 - 100 - 100 - 2$ with ReLU activations in all but the last one. We used the vectorized, $786$-dimensional version of MNIST as input to the parametric neighbor embedding methods (and not the $50$-dimensional PCA; but the s$k$NN graph was computed in the PCA space for consistency with non-parametric embeddings).

Like the reference NCVis implementation, we used the fractions $q_{\theta,Z}(x)/(q_{\theta,Z}(x) + m)$ instead of $q_{\theta,Z}(x)/\big(q_{\theta,Z}(x) + m\xi(x)\big)$. This is a mild approximation as the noise distribution is close to uniform. But it means that the model learns a scaled data distribution (cf. Cor. 2), so that we need to multiply the learned normalization parameter $Z$ by $n(n-1)$ when comparing to $t$-SNE or checking normalization of the NC-$t$-SNE model. Similarly, we also approximate the true noise distribution by the uniform distribution for the fractions $q_\theta(x)/(q_\theta(x) + \bar{Z}m/|X|)$ – instead of $q_\theta(x)/\big(q_\theta(x) + \bar{Z}m\xi(x)\big)$ – in our Neg-$t$-SNE implementation.

We mentioned in Sec. 5 and showed in Fig. S8 that one can move along the attraction-repulsion spectrum also by changing the number of noise samples $m$, instead of the fixed normalization constant $\bar{Z}$. In UMAP's reference implementation, there is a scalar prefactor $\gamma$ for the repulsive forces. Theoretically, adjusting $\gamma$ should also move along the attraction-repulsion spectrum, but setting it higher than $1$ led to convergence problems in (Böhm et al., 2022), Fig. A11. When varying our $\bar{Z}$, we did not have such issues.

For panels i in Figs. 1, S11 – S16 the Neg-$t$-SNE spectra were computed for $\bar{Z}$ equal to $Z(\theta^{t\text{-SNE}})$, $Z^{\text{NC-}t\text{-SNE}}$, and $\frac{n(n-1)}{m} \cdot x$, where $x \in \{5 \cdot 10^{-5}, 1 \cdot 10^{-4}, 2 \cdot 10^{-4}, 5 \cdot 10^{-4}, \dots, 1 \cdot 10^2, 2 \cdot 10^2, 5 \cdot 10^2\}$.

### K.4  SIMCLR EXPERIMENTS

For the SimCLR experiments, we trained the model for $1000$ epochs, of which we used $5$ epochs for warmup. The learning rate during warmup was linearly interpolated from $0$ to the initial learning rate. After the warmup epochs, we annealed the learning rate with a cosine schedule (without restarts) to $0$ (Loshchilov & Hutter, 2017). We optimized the model parameters with SGD and momentum $0.9$. We used the same data augmentations as in Chen et al. (2020) and their recommended batch size of $1024$. We used a ResNet18 (He et al., 2016) as the backbone and a projection head consisting of two linear layers ($512 - 1024 - 128$) with a ReLU activation in-between. The loss was applied to the $L_2$ normalized output of the projection head, but like Chen et al. (2020) we used the output of the ResNet as the representation for the linear evaluation. As similarity function we used the exponential of the normalized scalar product (cosine similarity) and always kept the temperature at $0.5$, as suggested in Chen et al. (2020).

Table S2: Run time overview for the most compute-heavy experiments

| | Number of runs | Time per run [min] (mean±SD) |
|---|---|---|
| Neg-$t$-SNE for Figs. 1, S11, S13, S14, S15 | 360 | $39 \pm 4$ |
| Neg-$t$-SNE for Fig. S16 | 24 | $121 \pm 44$ |
| Neg-$t$-SNE for Fig. S17 | 3 | $3592 \pm 44$ |
| NC-$t$-SNE (our implementation) for Fig. S10b | 3 | $786 \pm 2$ |
| SimCLR runs for Tab. 1 | 12 | $721 \pm 16$ |

Table S3: Learned normalization parameter for NC-$t$-SNE and partition function of $t$-SNE in our experiments. Mean and standard deviation is computed over three random seeds. In our setup $t$-SNE is deterministic.

| | $Z^{\text{NC-}t\text{-SNE}}$ $[10^6]$ | $Z(\theta^{t\text{-SNE}})$ $[10^6]$ |
|---|---|---|
| MNIST Fig. 1 | $34.3 \pm 0.1$ | 8.13 |
| MNIST without EE Fig. S11 | $34.3 \pm 0.1$ | 6.25 |
| MNIST imbalanced Fig. S12 | $6.15 \pm 0.06$ | 3.12 |
| Human brain organoid Fig. S13 | $3.57 \pm 0.03$ | 1.30 |
| Zebrafish Fig. S14 | $30.8 \pm 0.1$ | 7.98 |
| C. elegans Fig. S15 | $36.9 \pm 0.7$ | 11.7 |
| Kuzushiji-49 Fig. S16 | $395 \pm 3$ | 89.6 |

The ResNet was trained on the combined CIFAR-10 train and test sets. When evaluating the accuracy, we froze the backbone, trained the classifier on the train set, and evaluated its accuracy on the test set. We used sklearn's `KNeighborsClassifier` with cosine metric and $k = 15$ neighbors and sklearn's `LogisticRegression` classifier with the SAGA solver (Defazio et al., 2014), no regularization penalty, a tolerance of 0.0001, and `max_iter=1000`. Other parameters were left at the default values.

When sampling negative samples for a given head, we excluded that head from the candidates for negative samples. We sampled negative samples with replacement. If the desired number of negative samples $m$ equals twice the batch size minus 2 ($m = 2b - 2$, 'full-batch repulsion'), we took the entire batch without the current head and its tail as negative samples for that head.

### K.5 CODE AVAILABILITY

Our code is publicly available. The PyTorch implementation of contrastive neighbor embeddings can be found at `https://github.com/berenslab/contrastive-ne`. Details for reproducing the experiments and figures, alongside scripts and notebooks are at `https://github.com/hci-unihd/cl-tsne-umap`.

### K.6 STABILITY

Whenever we reported a metric or show a graph, we ran the experiments for 3 different random seeds and reported the mean $\pm$ the standard deviation. When the standard deviation was very small, we omitted it from the main text and report it in Tab. S3. Save for the usually approximate s$k$NN graph computation, $t$-SNE does not depend on a random seed. As we computed the s$k$NN exactly with PyKeOps (Charlier et al., 2021), $t$-SNE is deterministic in our framework.

Panels i in Figs. 1, S11 – S16 show the standard deviation as shaded area. Again, the standard deviations are very small and barely visible. The ratio of standard deviation to mean was never larger than 0.006 in panels i of Figs. 1, S11 – S16. Similarly, the standard deviation in Figs. S9 and S10, shown as shaded area, is mostly smaller than the line width.

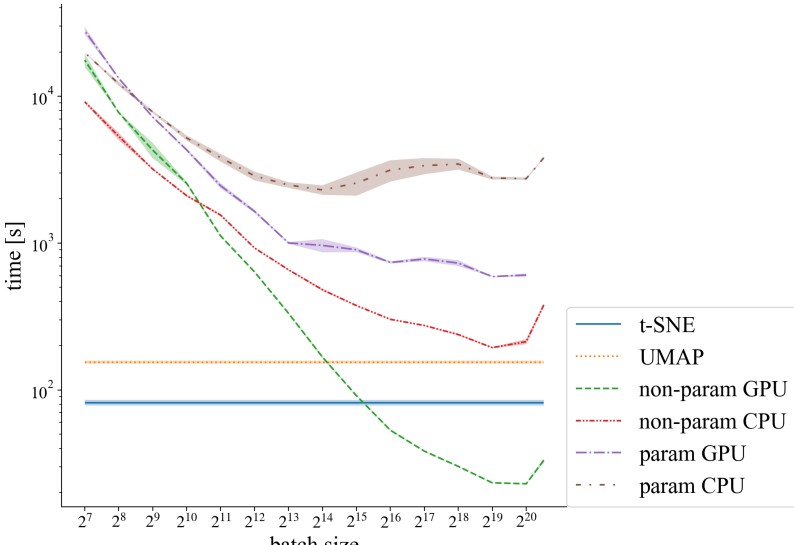

Figure S6: Run times of the embedding optimization phase for the MNIST dataset using different batch sizes, training modes, and hardware. Running on GPU is much faster than on CPU; non-parametric runs are faster than parametric runs. As the batch size increases, the run time decreases strongly in all settings.

### K.7 COMPUTE

We ran our neighbor embedding experiments on a machine with 56 Intel(R) Xeon(R) Gold 6132 CPU @ 2.60GHz, 502 GB RAM and 10 NVIDIA TITAN Xp GPUs. The SimCLR experiments were conducted on a SLURM cluster node with 8 cores of an Intel(R) Xeon(R) Gold 5220 CPU @ 2.20GHz and a Nvidia V100 GPU with a RAM limit of 54 GB. Each experiment used one GPU at most.

Our total compute is dominated by the neighbor embedding runs for Figs. 1, S11, S13–S16 and S10b and by the SimCLR experiments. Tab. S2 lists the number of runs and the average run time. We thus estimate the total compute time to be about 646 hours.

Our implementation relies on pure PyTorch and our experiments were conducted on GPUs. We originally kept the batch size for all our experiments at the default of 1024, motivated by Chen et al. (2020) and Sainburg et al. (2021), but eventually noticed that the run time of the visualization experiments depends crucially on the batch size (Fig. S6). Increasing the batch size to $2^{19}$ decreased the run time of the embedding optimization on MNIST data (without the s$k$NN graph computation) from about 40 min to just above 20 s. For a batch size of at least $2^{16}$, our implementation was faster than the reference implementations of UMAP and $t$-SNE (via openTSNE). The latter run only on CPU, while our experiments ran on GPU. Nevertheless, we observed a substantial improvement in run time for higher batch sizes also when running our PyTorch implementation on CPU and also when training a parametric embedding. Note that the parametric setting with full batch gradient descent (batch size 1,500,006 for MNIST's s$k$NN graph) exceeded our GPU's memory. On CPU, our implementation is just shy of the performance of UMAP when using $2^{19}$ attractive pairs per batch. Dedicated CUDA implementations (Chan et al., 2018; Eisenmann, 2019; Nolet et al., 2021) outperform our implementation on GPU. Compared to the Numba-accelerated CPU implementation of UMAP and the CUDA-accelerated GPU implementations, our pure PyTorch implementation arguably strikes a good speed / complexity trade-off, is easier to study and adapt by the machine learning community, and seamlessly integrates non-parametric and parametric settings as well as all four contrastive loss functions.

Note that the change from NC-$t$-SNE to Neg-$t$-SNE is as simple as fixing the learnable normalization parameter to a constant, so the original NCVis code written in C++ can easily be adapted to compute Neg-$t$-SNE. We have not used this for any of the experiments in our paper.

---

**Algorithm S1:** Batched contrastive neighbor embedding algorithm

---

**input** : list of directed s$k$NN graph edges $E = [i_1 j_1, \ldots, i_{|E|} j_{|E|}]$
    parameters $\theta$        // embeddings (non-param.)/ network weights
                              (param.)
    number of epochs $T$
    learning rate $\eta$
    number of noise samples $m$
    Cauchy kernel $q$   // of embeddings (non-param.)/ network output
                              (param.)
    batch size $b$
    loss mode $mode$
    normalization constant $\bar{Z}$                // default $\binom{n}{2}/m$, required for
                              $mode = \texttt{Neg-}t\texttt{-SNE}$

**output:** final embeddings $e_1, \ldots, e_n$

1 **if** $mode = NC\text{-}t\text{-}SNE$ **then**
2 $\quad$ $Z = 1$
3 **for** $t = 0$ **to** $T$ **do**
$\quad$ // Learning rate annealing
4 $\quad$ $\eta_t = \eta \cdot (1 - \frac{t}{T})$
5 $\quad$ $\alpha = 0$
6 $\quad$ **while** $\alpha < |E|$ **do**
7 $\quad\quad$ $\mathcal{L} = 0$
8 $\quad\quad$ **for** $\beta = 1, \ldots, b$ **do**
$\quad\quad\quad$ // Treat attractive edge $i_{\alpha+\beta} j_{\alpha+\beta}$ and noise edges $i_{\alpha+\beta} j_\mu^-$
$\quad\quad\quad$ // Sample noise edge tails but omit head of considered edge
9 $\quad\quad\quad$ $j_1^-, \ldots, j_m^- \sim \text{Uniform}(\{i_{\alpha+1}, j_{\alpha+1}, \ldots, j_{\alpha+b}\} \backslash \{i_{\alpha+\beta}\})$
$\quad\quad\quad$ // Aggregate loss based on mode
10 $\quad\quad\quad$ **if** $mode = Neg\text{-}t\text{-}SNE$ **then**
11 $\quad\quad\quad\quad$ $\mathcal{L} = \mathcal{L} - \log\left(\frac{q_\theta(i_{\alpha+\beta} j_{\alpha+\beta})}{q_\theta(i_{\alpha+\beta} j_{\alpha+\beta}) + \bar{Z} m / \binom{n}{2}}\right) - \sum_{\mu=1}^m \log\left(1 - \frac{q_\theta(i_{\alpha+\beta} j_\mu^-)}{q_\theta(i_{\alpha+\beta} j_\mu^-) + \bar{Z} m / \binom{n}{2}}\right)$
12 $\quad\quad\quad$ **else if** $mode = NC\text{-}t\text{-}SNE$ **then**
13 $\quad\quad\quad\quad$ $\mathcal{L} = \mathcal{L} - \log\left(\frac{q_\theta(i_{\alpha+\beta} j_{\alpha+\beta})/Z}{q_\theta(i_{\alpha+\beta} j_{\alpha+\beta})/Z + m}\right) - \sum_{\mu=1}^m \log\left(1 - \frac{q_\theta(i_{\alpha+\beta} j_\mu^-)/Z}{q_\theta(i_{\alpha+\beta} j_\mu^-)/Z + m}\right)$
14 $\quad\quad\quad$ **else if** $mode = InfoNC\text{-}t\text{-}SNE$ **then**
15 $\quad\quad\quad\quad$ $\mathcal{L} = \mathcal{L} - \log\left(\frac{q_\theta(i_{\alpha+\beta} j_{\alpha+\beta})}{q_\theta(i_{\alpha+\beta} j_{\alpha+\beta}) + \sum_{\mu=1}^m q_\theta(i_{\alpha+\beta} j_\mu^-)}\right)$
16 $\quad\quad\quad$ **else if** $mode = UMAP$ **then**
17 $\quad\quad\quad\quad$ $\mathcal{L} = \mathcal{L} - \log\left(q_\theta(i_{\alpha+\beta} j_{\alpha+\beta})\right) - \sum_{\mu=1}^m \log\left(1 - q_\theta(i_{\alpha+\beta} j_\mu^-)\right)$

$\quad\quad\quad$ // Update parameters with SGD (non-param.)  or Adam (param.)
18 $\quad\quad$ $\theta = \theta - \eta_t \cdot \nabla_\theta \mathcal{L}$
19 $\quad\quad$ **if** $mode = NC\text{-}t\text{-}SNE$ **then**
20 $\quad\quad\quad$ $Z = Z - \eta_t \nabla_Z \mathcal{L}$
21 $\quad\quad$ $\alpha = \alpha + b$
22 $\quad$ Shuffle $E$
23 **return** $\theta$

---

## L  ADDITIONAL FIGURES

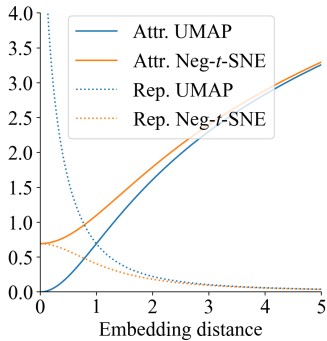

Figure S7: Attractive and repulsive loss terms of UMAP and Neg-$t$-SNE. The main difference is that UMAP's repulsive loss diverges at zero challenging its numerical optimization. The attractive terms are $\log(1 + d_{ij}^2)$ and $\log(2 + d_{ij}^2)$ for UMAP and Neg-$t$-SNE, respectively, and the repulsive ones are $\log\left((1 + d_{ij}^2)/d_{ij}^2\right)$ and $\log\left((2 + d_{ij}^2)/(1 + d_{ij}^2)\right)$, respectively.

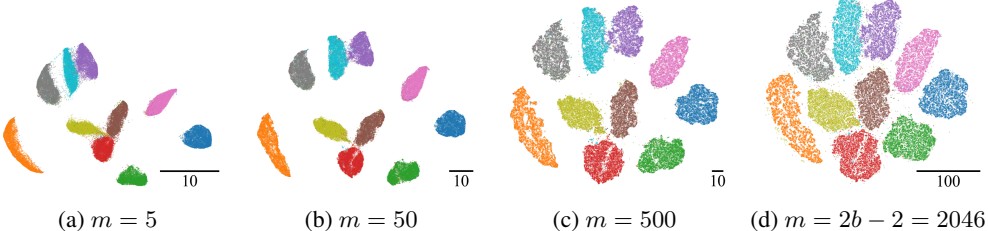

(a) $m = 5$      (b) $m = 50$      (c) $m = 500$      (d) $m = 2b - 2 = 2046$

Figure S8: Neg-$t$-SNE embeddings of the MNIST dataset for varying number of noise samples $m$ and using batch size $b = 1024$. While for NC-$t$-SNE and InfoNC-$t$-SNE more noise samples improve the approximation to $t$-SNE, see Figs. S18 and S19, changing $m$ in Neg-$t$-SNE moves the result along the attraction-repulsion spectrum (Fig. 1) with more repulsion for larger $m$. However, the computational complexity of Neg-$t$-SNE scales with $m$, so that moving along the spectrum via changing $\bar{Z}$ is much more efficient. For the first 250 epochs, $m$ was set to 5, to achieve an effect similar to early exaggeration (Supp. K).

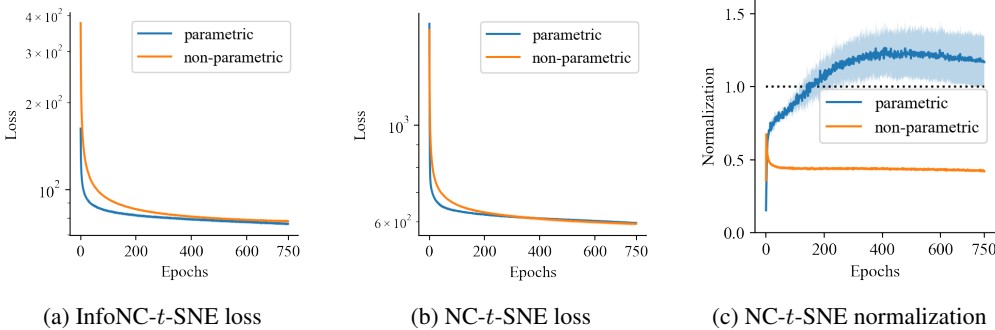

(a) InfoNC-$t$-SNE loss      (b) NC-$t$-SNE loss      (c) NC-$t$-SNE normalization

Figure S9: **(a, b)** Loss curves for the parametric and non-parametric InfoNC-$t$-SNE and NC-$t$-SNE optimizations leading to Figs. 3b, c, e, and f. While the embedding scale differs drastically between the non-parametric and the parametric run, the loss values are close. **(c)** Normalization of the model $\sum_{ij} \phi(ij)/Z$ for the parametric and non-parametric NC-$t$-SNE optimizations. The difference in the embedding scale is compensated by a three orders of magnitude change in $Z$, so that both versions learn approximately normalized models. These experiments used our NC-$t$-SNE reimplementation.

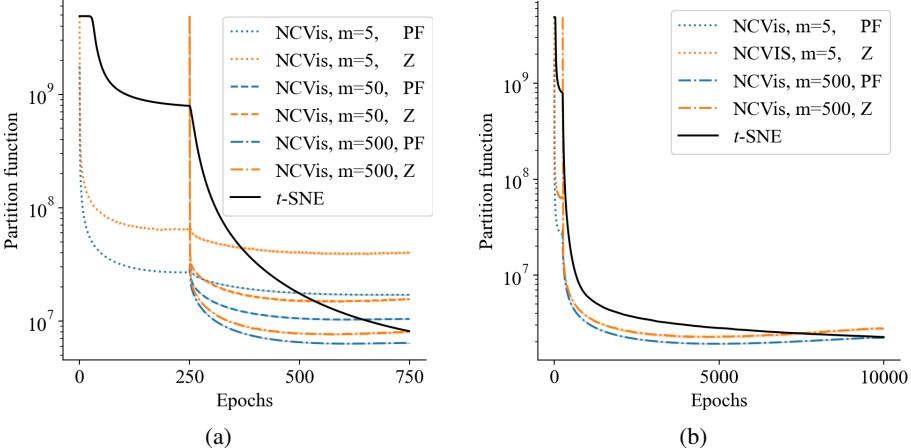

(a)      (b)

Figure S10: NC-$t$-SNE learns to have the same partition function (PF) as $t$-SNE on the MNIST dataset. The higher the number $m$ of noise samples **(a)** or the longer the optimization **(b)**, the better the match. Both methods used early exaggeration, which for NC-$t$-SNE meant to start with $m = 5$ noise samples for the first 250 epochs. The learned normalization parameter $Z$ converged to but did not exactly equal NC-$t$-SNE's partition function $\sum_{ij} q_\theta(ij)$. Nevertheless, it was of the same order of magnitude. Again, the match was better for more noise samples. Since we reinitialized the learnable $Z$ for NC-$t$-SNE after the early exaggeration phase, there were brief jumps in the partition function and in $Z$ at the beginning of the non-exaggerated phase.

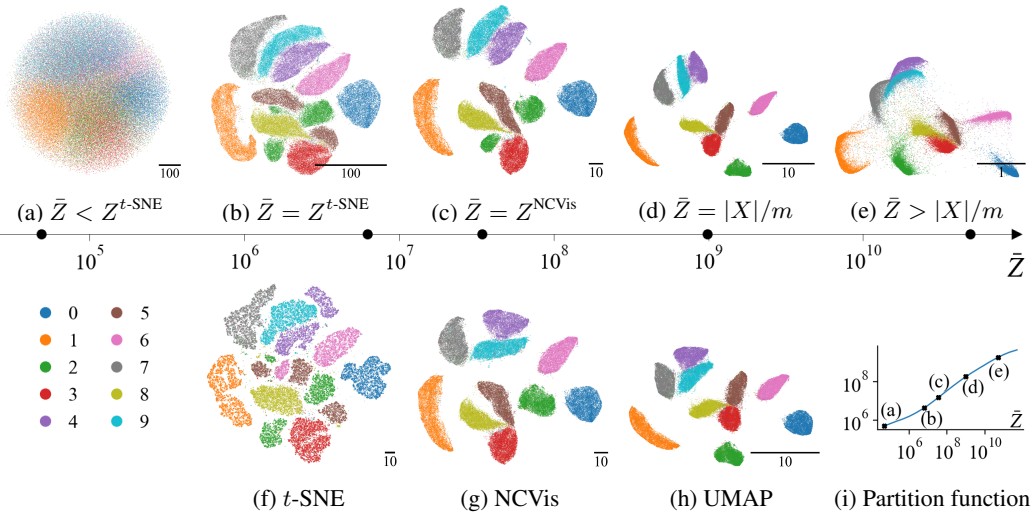

Figure S11: **(a–e)** Neg-$t$-SNE embeddings of the MNIST dataset for various values of the fixed normalization constant $\bar{Z}$. As $\bar{Z}$ increases, the scale of the embedding decreases, clusters become more compact and separated before eventually starting to merge. The Neg-$t$-SNE spectrum produces embeddings very similar to those of **(f)** $t$-SNE, **(g)** NCVis, and **(h)** UMAP, when $\bar{Z}$ equals the partition function of $t$-SNE, the learned normalization parameter $Z$ of NCVis, or $|X|/m = \binom{n}{2}/m$ used by UMAP, as predicted in Sec. 4–6. **(i)** The partition function $\sum_{ij}(1 + d_{ij}^2)^{-1}$ tries to match $\bar{Z}$ and grows with it. In contrast to Fig. 1, we did not use early exaggeration here, but initialized the Neg-$t$-SNE and $t$-SNE with PCA rescaled so that the first dimension has standard deviation 1 and 0.0001, respectively. This makes the embeddings with small $\bar{Z}$ values show cluster fragmentation, similar to the $t$-SNE embedding in **(f)** without early exaggeration. For very low $\bar{Z}$, the Neg-$t$-SNE embedding in **(a)** shows very little structure.

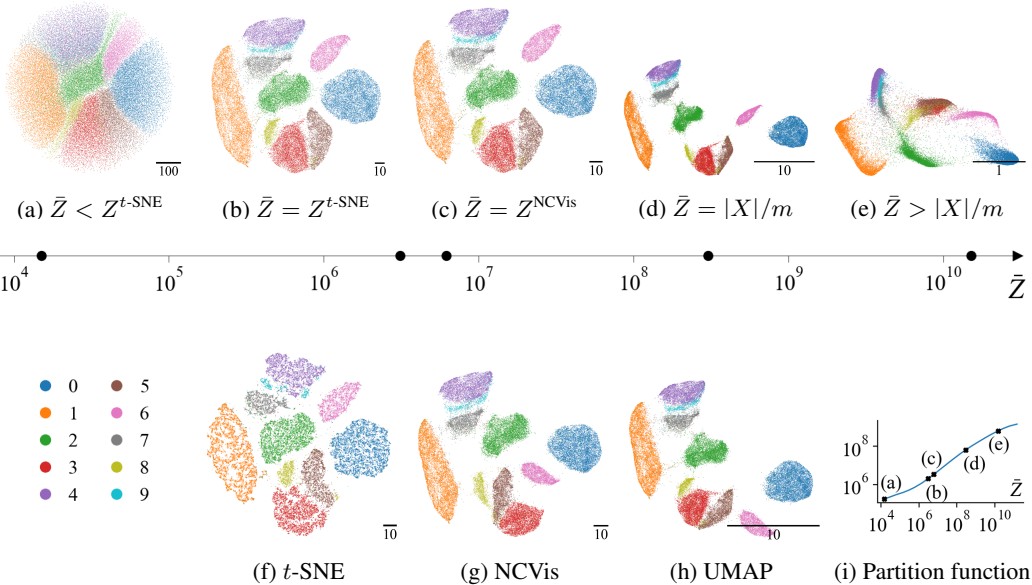

Figure S12: **(a–e)** Neg-$t$-SNE embeddings of an imbalanced version of the MNIST dataset for various values of the fixed normalization constant $\bar{Z}$. As $\bar{Z}$ increases, the scale of the embedding decreases, clusters become more compact and separated before eventually starting to merge. The Neg-$t$-SNE spectrum produces embeddings very similar to those of **(f)** $t$-SNE, **(g)** NCVis, and **(h)** UMAP, when $\bar{Z}$ equals the partition function of $t$-SNE, the learned normalization parameter $Z$ of NCVis, or $|X|/m = \binom{n}{2}/m$ used by UMAP, as predicted in Sec. 4–6. **(i)** The partition function $\sum_{ij}(1 + d_{ij}^2)^{-1}$ tries to match $\bar{Z}$ and grows with it. Similar to early exaggeration in $t$-SNE we started all Neg-$t$-SNE runs using $\bar{Z} = |X|/m$ and only switched to the desired $\bar{Z}$ for the last two thirds of the optimization. The dataset was created by randomly removing $10 \cdot c\%$ of the class of digit $c$, so that the class sizes linearly decrease from digit 0 to digit 9.

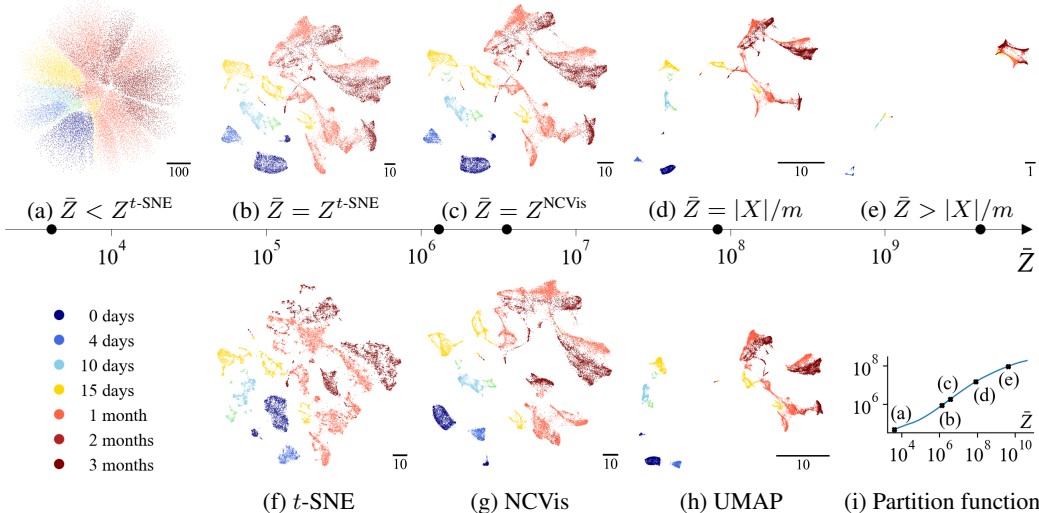

Figure S13: **(a – e)** Neg-$t$-SNE spectrum on the developmental single-cell RNA sequencing dataset from Kanton et al. (2019) for various parameters $\bar{Z}$. As $\bar{Z}$ increases, the scale of the embedding decreases and the continuous structure (corresponding to the developmental stage) becomes more apparent, making higher $\bar{Z}$ more suitable for visualizing continuous datasets (Böhm et al., 2022). The spectrum produces embeddings very similar to those of **(f)** $t$-SNE and **(g)** NCVis when $\bar{Z}$ equals the partition function of $t$-SNE or the learned normalization parameter of NCVis. The UMAP embedding in **(h)** closely resembles the Neg-$t$-SNE embedding at $\bar{Z} = |X|/m = \binom{n}{2}/m$. **(i)** The partition function $\sum_{ij}(1 + d_{ij}^2)^{-1}$ of the Neg-$t$-SNE embeddings increased with $\bar{Z}$. Similar to early exaggeration in $t$-SNE we started all Neg-$t$-SNE runs using $\bar{Z} = |X|/m$ and only switched to the desired $\bar{Z}$ for the last two thirds of the optimization. The dataset contains 20 272 cells and is colored by the duration of the development. There are ten times fewer cells collected after 10 days than after one month.

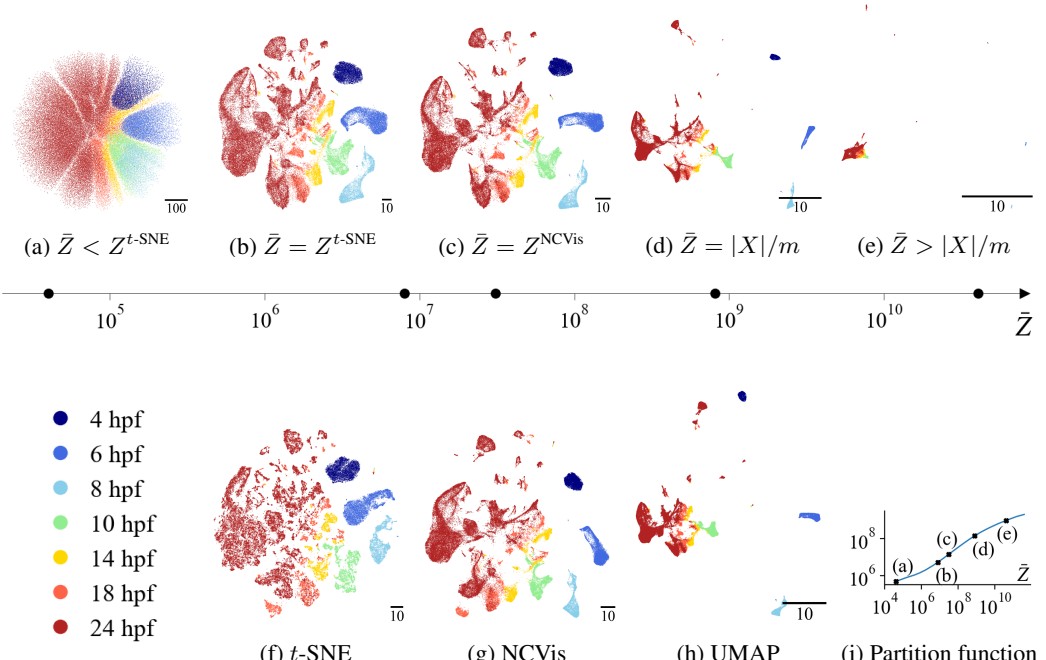

Figure S14: **(a – e)** Neg-$t$-SNE spectrum on the single-cell RNA sequencing dataset of a developing zebrafish embryo (Wagner et al., 2018) for various parameters $\bar{Z}$. As $\bar{Z}$ increases, the scale of the embedding decreases and the continuous structure (corresponding to the developmental stage) becomes more apparent, making higher $\bar{Z}$ more suitable for visualizing continuous datasets (Böhm et al., 2022). The spectrum produces embeddings very similar to those of **(f)** $t$-SNE and **(g)** NCVis when $\bar{Z}$ equals the partition function of $t$-SNE or the learned normalization parameter of NCVis. The UMAP embedding in **(h)** closely resembles the Neg-$t$-SNE embedding at $\bar{Z} = |X|/m = \binom{n}{2}/m$. **(i)** The partition function $\sum_{ij}(1 + d_{ij}^2)^{-1}$ of the Neg-$t$-SNE embeddings increased with $\bar{Z}$. Similar to early exaggeration in $t$-SNE we started all Neg-$t$-SNE runs using $\bar{Z} = |X|/m$ and only switched to the desired $\bar{Z}$ for the last two thirds of the optimization. The dataset contains $63\,530$ cells and is colored by the hours post fertilization (hpf). There are ten times fewer cells collected after $8$ hours than after $24$.

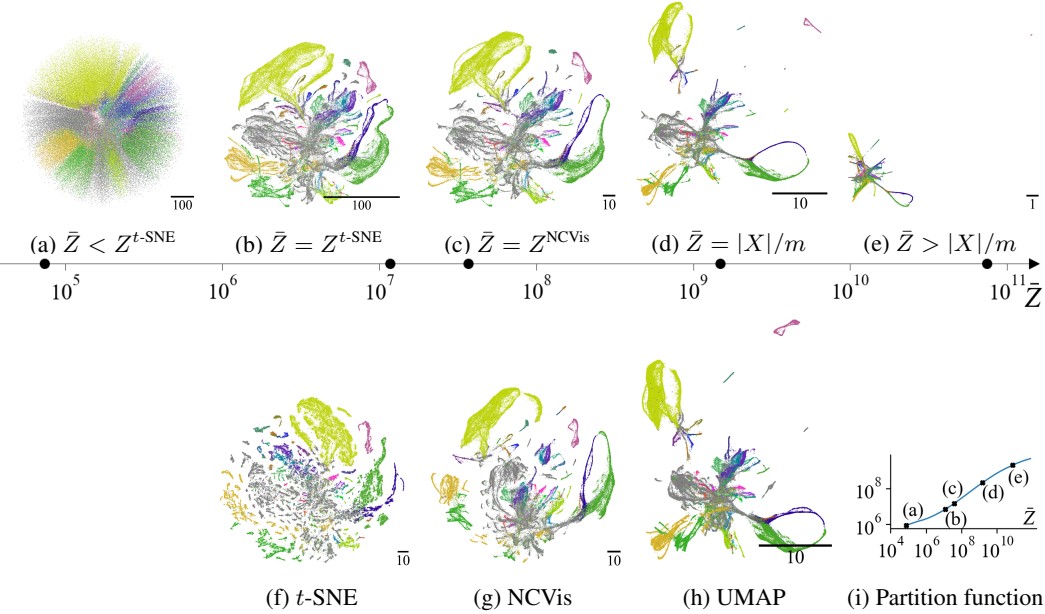

Figure S15: **(a–e)** Neg-$t$-SNE spectrum of the single-cell RNA sequencing dataset of the C. elegans flatworm (Packer et al., 2019; Narayan et al., 2021) for various values of the fixed normalization constant $\bar{Z}$. As $\bar{Z}$ increases, the scale of the embedding decreases, the scale of the embedding decreases, and more continuous structure becomes apparent. The Neg-$t$-SNE spectrum produces embeddings very similar to those of **(f)** $t$-SNE, **(g)** NCVis, and **(h)** UMAP, when $\bar{Z}$ equals the partition function of $t$-SNE, the learned normalization parameter $Z$ of NCVis, or $|X|/m = \binom{n}{2}/m$ used by UMAP, as predicted in Sec. 4–6. **(i)** The partition function $\sum_{ij}(1 + d_{ij}^2)^{-1}$ tries to match $\bar{Z}$ and grows with it. Similar to early exaggeration in $t$-SNE we started all Neg-$t$-SNE runs using $\bar{Z} = |X|/m$ and only switched to the desired $\bar{Z}$ for the last two thirds of the optimization. The dataset contains information on $86\,024$ cells of 37 types indicated by the colors. It is imbalanced with only 25 cells of the least abundant type but $31\,375$ cells of unknown type (grey).

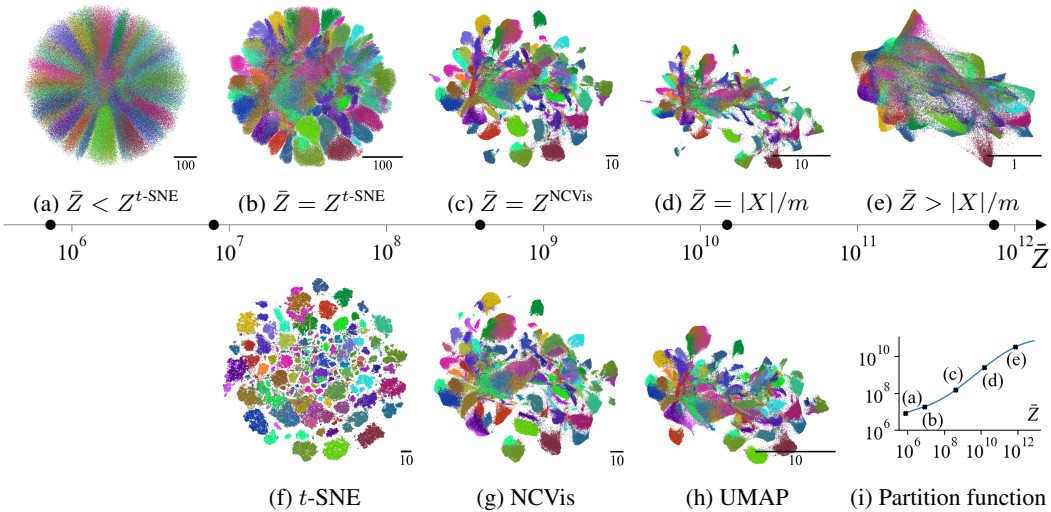

(f) $t$-SNE $\qquad$ (g) NCVis $\qquad$ (h) UMAP $\qquad$ (i) Partition function

Figure S16: **(a − e)** Neg-$t$-SNE embeddings of the Kuzushiji-49 dataset (Tarin et al., 2018) for various values of the fixed normalization constant $\bar{Z}$. As $\bar{Z}$ increases, the scale of the embedding decreases, clusters become more compact and separated before eventually starting to merge. The Neg-$t$-SNE spectrum produces embeddings similar to those of **(f)** $t$-SNE, **(g)** NCVis, and **(h)** UMAP, when $\bar{Z}$ equals the partition function of $t$-SNE, the learned normalization parameter $Z$ of NCVis, or $|X|/m = \binom{n}{2}/m$ used by UMAP, as predicted in Sec. 4–6. **(i)** The partition function $\sum_{ij}(1+d_{ij}^2)^{-1}$ tries to match $\bar{Z}$ and grows with it. Similar to early exaggeration in $t$-SNE we started all Neg-$t$-SNE runs using $\bar{Z} = |X|/m$ and only switched to the desired $\bar{Z}$ for the last two thirds of the optimization. The dataset contains $270\,912$ images of 49 different Japanese characters. The classes are imbalanced with 456 to $7\,000$ samples per class. We see that a higher level of repulsion than UMAP's $\bar{Z} = |X|/m$ helps to visualize the discrete structure of the dataset. The sampling based Neg-$t$-SNE embedding at $\bar{Z} = Z^{t\text{-SNE}}$ has less structure than the $t$-SNE embedding. Fig. S17 and Tab. S4 show that the Neg-$t$-SNE result improves for longer optimization.

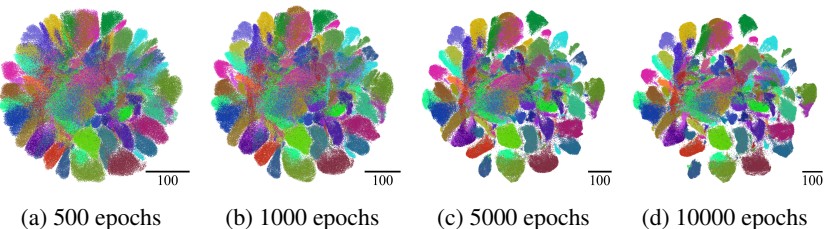

(a) 500 epochs $\qquad$ (b) 1000 epochs $\qquad$ (c) 5000 epochs $\qquad$ (d) 10000 epochs

Figure S17: Neg-$t$-SNE embeddings of the Kuzushiji-49 dataset (Tarin et al., 2018) for $\bar{Z} = Z^{t\text{-SNE}}$ show more structure when optimized longer. Similar to early exaggeration in $t$-SNE we started all Neg-$t$-SNE runs using $\bar{Z} = |X|/m$ for 250 epochs and only switched to the desired $\bar{Z}$ for the remaining number of epochs indicated in the subcaptions.

Table S4: Longer run times improve the Neg-$t$-SNE optimization on the Kuzushiji-49 dataset (Tarin et al., 2018) for $\bar{Z} = Z^{t\text{-SNE}}$. The KL divergence is computed with respect to the normalized model $q_\theta/(\sum_{ij} q_\theta(ij))$.

| Epochs | 500 | 1000 | 5000 | 10000 |
|---|---|---|---|---|
| Partition function [$10^6$] | $18.96 \pm 0.02$ | $13.27 \pm 0.01$ | $6.93 \pm 0.01$ | $5.75 \pm 0.01$ |
| Neg-$t$-SNE loss | $1021 \pm 2$ | $832 \pm 2$ | $630 \pm 1$ | $599 \pm 1$ |
| KL divergence | $5.52 \pm 0.01$ | $5.18 \pm 0.01$ | $4.76 \pm 0.01$ | $4.65 \pm 0.00$ |

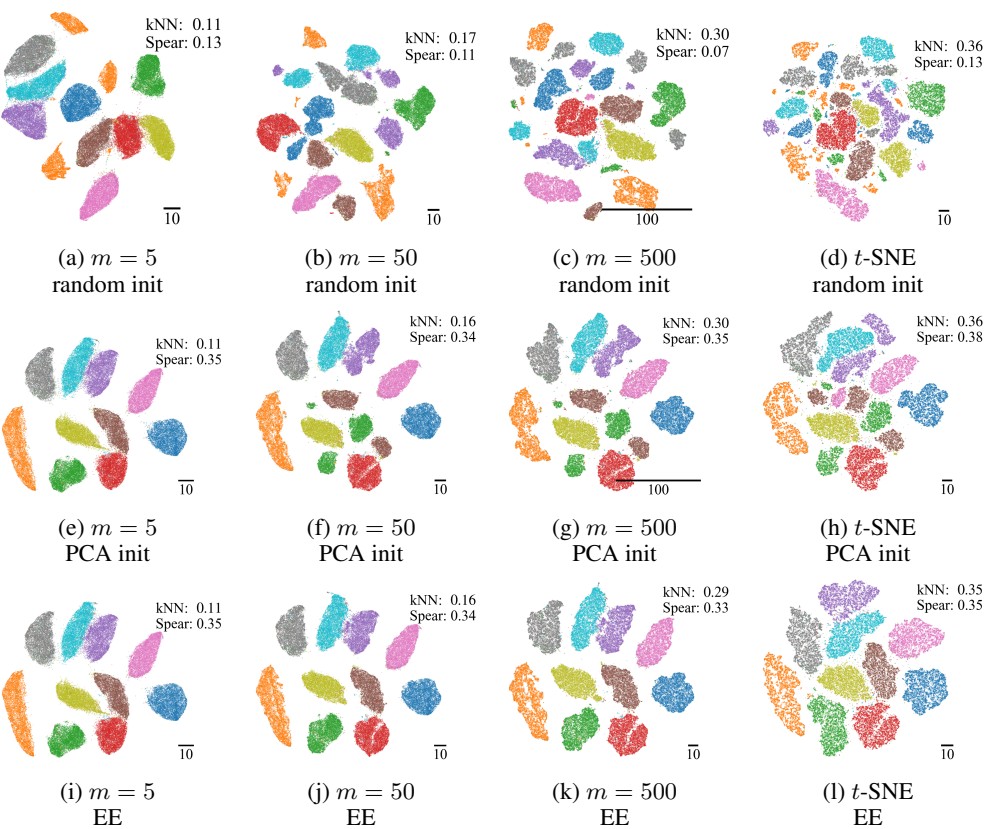

Figure S18: NC-$t$-SNE (our implementation) on the MNIST dataset for varying number of noise samples $m$ and different starting conditions. Higher number of noise samples $m$ improves the approximation quality to $t$-SNE (last column). The first row is initialized with isotropic Gaussian noise and the second and the third rows with PCA (both normalized to have standard deviation of one or 0.0001 in the first dimension for NC-$t$-SNE or $t$-SNE, respectively). In the third row, the first 250 epochs used $m = 5$ and the latter used the given $m$ value for NC-$t$-SNE. This is similar to $t$-SNE's early exaggeration that we used in panel l. NC-$t$-SNE seems to be less dependent on early exaggeration than $t$-SNE, especially for low $m$ values. Insets show the $k$NN recall and the Spearman correlation between distances of pairs of points. Higher $m$ increases the $k$NN recall, while mostly leaving the Spearman correlation unchanged. Random initialization hurts the Spearman correlation, but not the $k$NN recall.

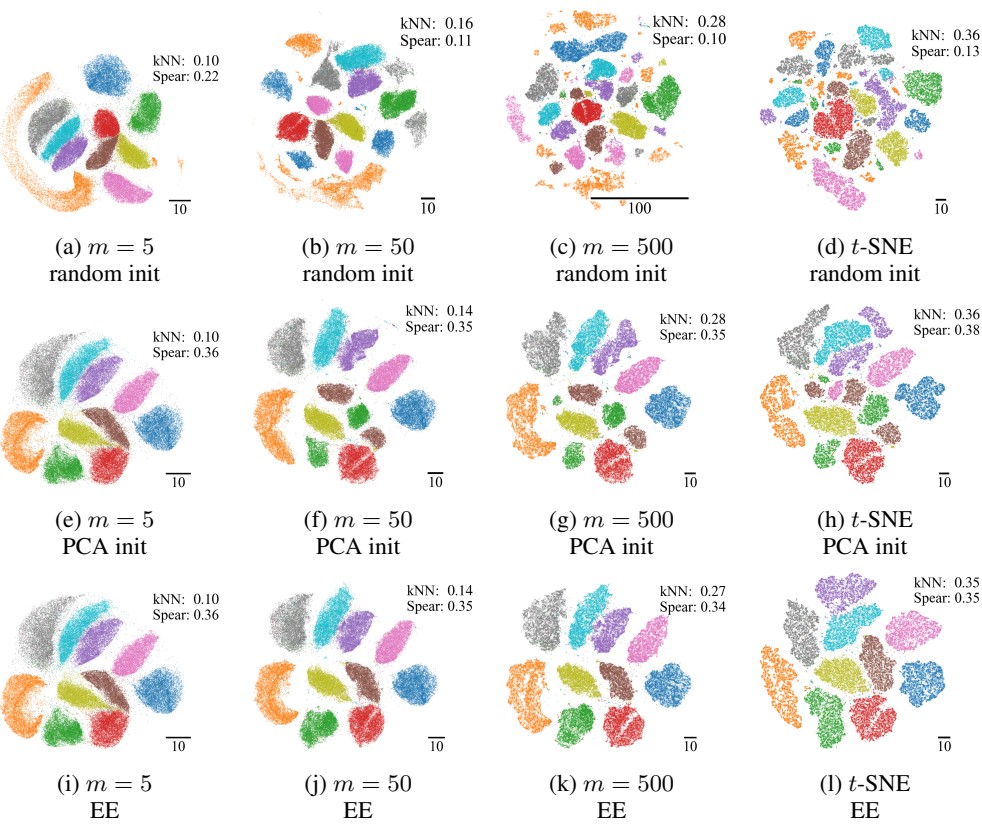

Figure S19: InfoNC-$t$-SNE on the MNIST dataset for varying number of noise samples $m$ and different starting conditions. Higher number of noise samples $m$ improves the approximation quality to $t$-SNE (last column). The first row is initialized with isotropic Gaussian noise and the second and the third rows with PCA (both normalized to have standard deviation of one or $0.0001$ in the first dimension for InfoNC-$t$-SNE or $t$-SNE, respectively). In the third row, the first 250 epochs used $m = 5$ and the latter used the given $m$ value for InfoNC-$t$-SNE. This is similar to $t$-SNE's early exaggeration that we used in panel l. InfoNC-$t$-SNE seems to be less dependent on early exaggeration than $t$-SNE, especially for low $m$ values. Insets show the $k$NN recall and the Spearman correlation between distances of pairs of points. Higher $m$ increases the $k$NN recall, while mostly leaving the Spearman correlation unchanged. Random initialization hurts the Spearman correlation, but not the $k$NN recall.

