# OpenReview forum: "From $t$-SNE to UMAP with contrastive learning"
_ICLR.cc/2023/Conference — ICLR 2023 poster_

### Official Review · Reviewer_ZbpN · 2022-10-22

**Confidence:** 3
**Correctness:** 4
**Technical Novelty And Significance:** 4
**Empirical Novelty And Significance:** 3
**Recommendation:** 8

**Clarity, Quality, Novelty And Reproducibility:**

- the paper is clearly written and of high quality
- to my knowledge the core relationship of NCE & NEG and its relation to UMAP & t-SNE is novel
- a software package is published alongside for reproducability


**Strength And Weaknesses:**

Strenghts.
- good presentation of a novel insight connecting NCE and NEG
- well and clearly written paper
- new insight about a connection of t-SNE and UMAP, which spans a range of methods

Weaknesses.
- the relation of NCE and NEG could benefit from more empirical support. In particular, Corollary 2 states that the resulting function q is proportional to the density with \hat Z. Fig1(i) shows a monotonic trend, but not a precise relation (as argued on the text). For a simpler model I'd expect that one could be able to establish this relationship, which would empirically harden the statement.
- Equation 10 shows that a Cauchy Kernel in UMAP corresponds to a inverse-square kernel in NEG. However, as far as I understood, you're using a Cauchy kernel in NEG. It'd be illustrative to see the comparison of NEG and explicit inverse-square kernel - which should show a closer resemblance to UMAP. Also, I think this difference in kernels when interpolating between the methods should be emphasized when talking about the interpolation ability.

**Summary Of The Paper:**

The paper "From t-SNE to UMAP with contrastive learning" presents a new insight into the connection of two contrastive learning methods: noise contrastive estimation (NCE) and negative sampling (NEG). This insight allows to connect t-SNE and UMAP, and span a spectrum of methods along their connection. Finally, connection to other self-supervised methods (such as SimCLR) are established. Figures of the method show dimensionality reductions on MNIST. The method is accompanied by a package in PyTorch.

**Summary Of The Review:**

The paper is well written and offers novel insights into the relation of NCE & NEG. This paves the way to interpolate between UMAP and t-SNE, two highly used dimensionality reduction techniques.

---

> ### Author Response · Authors · 2022-11-17
> **Reply to reviewer ZbpN**
>
> We thank reviewer ZbpN for the time invested in reviewing our submission and for the positive feedback!
>
> **Empirical support for the relation of NCE and NEG:**
>
> It is true that for the MNIST dataset in Fig. 1, we only observed a monotonic relationship between $\bar{Z}$ and the resulting partition function, instead of an identity as predicted by our Cor. 2. This monotone relation persists over all the datasets in Figs. S10 - S15, panel (i), and is close to linear in the reasonable range of $\bar{Z}$, between $t$-SNE's partition function and UMAP's normalization constant. As you mention, we discuss reasons for this discrepancy of the fit in Sec. 5 and the Discussion.
>
> That said, following your suggestion, we have now run Neg-$t$-SNE on a toy dataset consisting of three equidistant points. Indeed, we found that in this simplified setting the partition function of the Neg-$t$-SNE embedding exactly matched $\bar{Z}$, in agreement with the theory. We included these results in a new appendix I (Figure S5) and mention it in the Discussion. Thank you for suggesting this experiment!
>
>
> **Inverse square kernel in NEG:**
> We rephrased the key point of Sec. 6 into the new Lem. 3 which shows that UMAP's loss is precisely NEG with inverse square kernel. Thus, any implementation of UMAP uses NEG with inverse square kernel (in our terminology, one could call it Neg-$1/d^2$-SNE). Differences between UMAP's reference implementation and combining the inverse square kernel with our implementation of NEG are thus only in details such as the size of mini-batches, explicit implementation of the gradient versus reliance on PyTorch's autodiff, etc. In fact, our submission contains UMAP plots based on both implementations: In Figs. 1 h) and S1 we use UMAP's reference implementation (with PCA init and binary weights), while in Figs. 2 a), b) and S2 we use our PyTorch framework to reimplement UMAP. This differs from Neg-$t$-SNE only in the choice of kernel. We observe that both in our reimplementation of UMAP as well as the reference implementation of UMAP, the visual effect of using the inverse square kernel is mitigated by the strong learning rate annealing to zero. Switching it off yields a fuzzy pattern both in our reimplementation and in the reference implementation (which we extended to toggle learning rate annealing), see Figs. 2 a), b), S1 and S2. The Neg-$t$-SNE embedding in Fig. 2 c), d) is much more robust against toggling the learning rate annealing.
>
> This confirms your conjecture that combining NEG with the inverse square kernel yields a closer fit to UMAP than Neg-t-SNE. However, this difference becomes only apparent when switching off the learning rate annealing.

---

### Official Review · Reviewer_gEqf · 2022-10-23

**Confidence:** 2
**Correctness:** 3
**Technical Novelty And Significance:** 2
**Empirical Novelty And Significance:** 2
**Recommendation:** 5

**Clarity, Quality, Novelty And Reproducibility:**

The write-up is clear, but the main contributions of the paper is not highlighted and distinguished from previous work.

**Strength And Weaknesses:**

### Strength:
* The authors do a reasonably well job motivating the problem and discussing NCE and NEG in the context of word embedding and dimensionality reduction.

### Weaknesses:
* The current version of the paper fails to distinguish itself from previous work, especially different variants of t-SNE (e.g., Barnes-Hut t-SNE) or its relation to UMAP, discussed in Böhm et al. (2022). It is hard to pinpoint the main contribution of the paper ito using negative samples (already done in UMAP and LargeVis).
* The main shortcoming is the lack of comparison to TriMap, which is the only DR method that truly relies on contrastive learning! Unlike what the authors discuss in Section 2, TriMap is not a variant of UMAP (TriMap stands for Triplet **Mapping**, IIUC, not Manifold Approximation and Projection) and uses NEG sampling (i.e., triplets) to form an embedding. It is well known that TriMap provides a distinct view of the data ito the global structure. I find the lack of comparison and discussion to TriMap as the main shortcoming of the paper.

**Summary Of The Paper:**

The authors discuss noise contrastive estimation (NCE) and negative sampling (NEG) in the context of dimensionality reduction and their connection to t-SNE and UMAP. They propose a variant of t-SNE, called NEG-t-SNE, which relies on NEG to reduce the complexity and produce more compact clusters.

**Summary Of The Review:**

The authors need to expand the connection to contrastive learning and, especially TriMap and provide comparisons. They also need to highlight their main contributions and distinguish the new results from previous approaches such as LargeVis. Also, I recommend providing further evidence that tuning other hyperparameters in t-SNE and UMAP (such as perplexity) fails to provide a similar effect as the one proposed in NEG-t-SNE.

---

> ### Author Response · Authors · 2022-11-16
> **Reply to reviewer gEqf (1/2)**
>
> Dear reviewer gEqf,
> thank you very much for the time invested into reading our paper and the detailed feedback you provided! In the revised manuscript, we have added a detailed discussion of TriMap and other contrastive neighbour embedding methods.
>
> **TriMap**
> Thank you for emphasizing the relation of TriMap to our work. We have improved the description of TriMap in Sec. 2 and added 1.5 pages discussing TriMap's relation to InfoNC-$t$-SNE in the new Suppl. F "TriMap and InfoNC-$t$-SNE". Our main finding is that after stripping away several bells and whistles from TriMap, it turns out to be similar to our InfoNC-$t$-SNE in the special case of using only $m=1$ noise sample.
>
> We also found that using our default of $m=5$ noise samples improves both a local and a global metric compared to default TriMap.
>
> Please note that in the manuscript we use the term "negative sampling" in the strict sense of Mikolov et al. 2013, as described in our Eq. (8). We are aware that "negative sampling" is sometimes used in a loose sense meaning the use of any form of contrastive learning involving "negative samples". TriMap only qualifies as "negative sampling" in the latter, loose sense and not in the way we use the term in the manuscript. Also, we disagree with your statement that TriMap is "the only DR method that truly relies on contrastive learning". For example, NCVis, which is based on noise-*contrastive*-estimation, as well as UMAP, which uses negative sampling (in the strict sense, see our Sec. 6), are also contrastive dimension reduction methods.
>
> **Differences to Böhm et al. 2022:**
> While both our work and Böhm et al. aim to connect $t$-SNE and UMAP, our approach is much more theoretically grounded and achieves a precise connection. In particular, Böhm et al. did not work with UMAP's true loss function in closed form (Damrich et al. 2021), which is integral for understanding the conceptual relation between $t$-SNE and UMAP. Furthermore, they did not realize the connection between UMAP's use of NEG and the approximation of $t$-SNE with NCE (our Sec. 4). Finally, our further novel insight is that UMAP, when understood as negative sampling _sensu stricto_, uses the $1/d^2$ kernel instead of the Cauchy kernel and hence requires learning rate annealing.
>
> Note that the main point of our work is not to suggest using NEG for DR, but to show how NEG, and thus the methods using it, relate to other DR approaches like NCVis and $t$-SNE.
> The closest that our work gets to Böhm et al. is in the negative sampling spectrum in Sec. 5. We conclude that section with a discussion of the relation to Böhm et al. Therefore, we feel that we have already adequately discussed the relationship between Böhm et al. and our work.

---

> > ### Author Response · Authors · 2022-11-16
> > **Reply to reviewer gEqf (2/2)**
> >
> > **Tuning other hyperparameters in $t$-SNE and UMAP:**
> > As we describe in Sec. 3.2, neighbor embedding methods, like $t$-SNE and UMAP, can be decomposed into separate ingredients, such as high-dimensional similarity, initialization, low-dimensional similarity kerne, and loss function. Prior work (Böhm et al. 2022 and Damrich et al. 2021) showed that both $t$-SNE and UMAP can be made to use the exact same high-dimensional similarities and the low-dimensional similarity kernel, without qualitative changes to the two embeddings, i.e., $t$-SNE and UMAP remain different even if these two ingredients are set the same. While the choice of initialization is important (Kobak et al. 2021), using identical initialization does not turn the characteristic look of a $t$-SNE embedding into that of a UMAP embedding either. Therefore, the main difference between t-SNE and UMAP lies in their loss functions, and our work relates them conceptually for the first time. Note that, unless otherwise stated, we always used the same high-dimensional similarity, the same initialization, and the same low-dimensional similarity kernel in all our experiments. Only the loss functions differ.
> >
> > We do not claim that it is not possible to change some hyperparameters of $t$-SNE and UMAP to overcome the described difference in their loss functions (even though we do find it unlikely). But this would only be an ad-hoc fix. Let us elaborate with the example of $t$-SNE's perplexity. By default, $t$-SNE and UMAP effectively consider about 15 nearest neighbors. Increasing $t$-SNE's perplexity amounts to considering more nearest neighbors, which indeed changes the layout. Perhaps in some cases this could lead to a more UMAP-like embedding, but then a difference in the number of nearest neighbors would effectively cancel out the discrepancy in the loss functions. Instead, we tackle the root of $t$-SNE's and UMAP's difference and propose a way to provably interpolate their loss functions (and their gradients, see Suppl. B).
> >
> > **Other $t$-SNE variants, LargeVis and PaCMAP:**
> > A comprehensive review of the visualization methods of the last decade is beyond the scope of our paper. We focus on connecting the dots between the two most prominent examples, $t$-SNE and UMAP. Nevertheless, we have now included brief discussions of several other methods in a new Suppl. G "Relation to other visualization methods". We discuss the BH-$t$-SNE (van Maaten et al. 2014) and FI$t$-SNE (Linderman et al. 2019) approximations to $t$-SNE and mention their worse scaling with respect to the embedding dimension compared to the contrastive neighbor embeddings in our work. In addition, we highlight that our analysis of UMAP largely transfers to the similar method LargeVis (Tang et al. 2016) and also relate the loss of PaCMAP (Wang et al. 2021) to that of Neg-$t$-SNE.

---

### Official Review · Reviewer_xsMD · 2022-10-24

**Confidence:** 3
**Correctness:** 3
**Technical Novelty And Significance:** 3
**Empirical Novelty And Significance:** Not applicable
**Recommendation:** 8

**Clarity, Quality, Novelty And Reproducibility:**

The paper is well-written. The paper has good quality and is very novel, although I do not check the correctness of all the theory.

**Strength And Weaknesses:**

Pros:

1. The paper provides very novel findings. Its theory (Corollary 2) explains the NEG’s behavior in the minibatch setting. The theory seems to predict the fact that Neg-t-SNE produced more compact clusters than t-SNE. Based on its theory, the paper reveals that UMAP is NEG applied to the t-SNE framework. It explains the empirical phenomenon that UMAP pulls embedding points closer together than t-SNE.

2. The paper is well-written and well-structured. I am not an expert in the field. I am not able to verify the correctness of the theory. However, following the explanation of the paper. I can grasp the high-level idea and understand the relationship between UMAP and t-SNE.

Suggestions:

1. It would be better if the author can give further instructions to the practitioner about how to better use UMAP and t-SNE in real practice according to the findings of this paper. As the author says, "Our conceptual relation between UMAP and t-SNE is therefore of practical impact – as is the fact that both are just instances on a spectrum highlighting more continuous or more discrete structures." How should this fact affect our daily usage of UMAP and t-SNE.

**Summary Of The Paper:**

The paper reveals the connection between t-SNE and UMAP via a new connection between the contrastive methods NCE and NEG. The paper also builds the connection between contrastive neighborhood embedding and self-supervised contrastive learning.

**Summary Of The Review:**

Overall, I am very excited about this work. As an ML practitioner, I use t-SNE and UMAP almost everyday. I always wonder what is the connection between them. This paper answers my question. I think this paper's findings are useful to the majority of the ML community.

---

> ### Author Response · Authors · 2022-11-16
> **Reply to reviewer xsMD**
>
> Dear reviewer xsMD,
> thank you for taking the time to read our manuscript and for your positive feedback!
>
> **Practical suggestions:**
> Our work shows that $t$-SNE and UMAP are instances on a spectrum of embedding methods, parametrized with $\bar{Z}$, which yield insightful embeddings over a large range of $\bar{Z}$. Rather than finding an "optimal" place on this spectrum, we believe it makes sense to offer a "slider" to allow practitioners to tune attraction and repulsion and observe how structure arises or decays when sliding through the spectrum. We believe that this will make practitioners less likely to be misled by artifacts of any given embedding.
>
> As described in Sec.8, exploration of the spectrum does not require specialized knowledge. For UMAP, $\bar{Z}^{\text{UMAP}} = \binom{n}{2} / m$; for $t$-SNE, Böhm et al. 2022 found that the partition function $Z^{t\text{-SNE}}$ typically lies in $[50n, 100n]$. Both quantities can be computed a priori. We extended our PyTorch package so that it allows the user to move along the spectrum with a slider parameter $s$ such that $s=0, 1$ corresponds to $\bar{Z} = Z^{t\text{-SNE}}, \bar{Z}^{\text{UMAP}}$, respectively, without any need for specifying $\bar{Z}$ directly.
>
> As dimensionality reduction necessarily loses information, we feel that providing practitioners with a larger toolbox allows them to more faithfully explore their datasets. Moreover, already the knowledge that $t$-SNE and UMAP are only points on a spectrum of embedding methods might caution against over-interpretation of one or the other.
>
> If prior information on the dataset structure is available, certain regimes of the spectrum might be more useful than others. Higher attraction, on the one hand, allows for more distant parts of the embedding to attract, which can bring out global structure. On the other hand, this can lead to local-overcontraction (Damrich et al. 2021). Thus, if the experimentalist expects the data to have prominent global structure, e.g., because the data comes from a developmental process, we argue for more attraction. As examples of such datasets, consider Figs. S12 and S13 which show developmental single-cell RNA sequencing datasets. The main feature, the temporal trajectory, is visible in S12 (d), where we employ high attraction (high $\bar{Z}$). However, many regions are squeezed almost to one-dimensional lines or tiny clusters (one-month and zero-day cells). Increasing the repulsion (decreasing $\bar{Z}$) leads to an embedding in S12 (b), where we see much richer local structure in previously over-contracted regions. Unfortunately, the global trajectory is no longer visible.
>
> Conversely, if the dataset is not expected to contain relevant global structure, more repulsion can help to achieve better local resolution. Staying with the single-cell example, this might be the case if the data come from an adult and fully differentiated cell population, in which one only expects to see separated clusters. Alternatively, our Fig. S15 shows a visualizations of a dataset consisting of images of 49 Japanese characters, where one would just expect to see those 49 clusters. High attraction, as in panels (d) and (e), falsely merges different clusters, while higher repulsion, like in panels (b) or (f) separates the clusters better.
>
> We have extended the discussion to emphasize these points.

---

### Official Review · Reviewer_fZKZ · 2022-10-24

**Confidence:** 4
**Correctness:** 2
**Technical Novelty And Significance:** 2
**Empirical Novelty And Significance:** 2
**Recommendation:** 3

**Clarity, Quality, Novelty And Reproducibility:**

I do not find this paper very clearly written. The overall argument made about the normalization constant $\bar Z$ is plausible, but not overwhelmingly convincing. Many statements in Section 3 and 4 may be partly known in the literature, so they may not be novel. I did not check if the results are reproducible or not.

**Strength And Weaknesses:**

**Strengths**: it is quite useful to compare and analyze the two popular data visualization techniques, namely t-SNE and UMAP. I find it plausible that the normalization constant $\bar Z$ is playing a crucial role in controlling the compactness of embedded clusters.

**Weaknesses**: In my opinion, most arguments made in this paper are quite casual and handwaving. Other than the role of $\bar Z$, I do not see a clear argument being made, so I didn't find myself more informed after reading this paper.

There are also crucial parts of t-SNE and UMAP that are not carefully discussed. For example, it is known that using random initialization or spectral initialization has a great impact on the visualization quality. Also, there are parameters in UMAP that determine how fat the tail is in the loss function, which presumably controls the compactness of embedded clusters.

**Summary Of The Paper:**

The authors propose to unify two popular neighborhood embedding method, t-SNE and UMAP, from the perspective of contrastive learning. It is shown that a single normalization constant $\bar Z$ is controlling the visualization quality, and the same constant interpolates between t-SNE and UMAP. By making the connections, the author argue that the instability of UMAP can be remedied.

**Summary Of The Review:**

The authors make connections between t-SNE and UMAP from the contrastive learning perspective, notably using a normalization constant $\bar Z$. I find that many statements are quite vague and informal, which unfortunately makes the contribution unclear and obscure. I would not recommend acceptance of this paper.

---

> ### Author Response · Authors · 2022-11-16
> **Reply to reviewer fZKZ (1/2)**
>
> Thank you for the time and effort invested into reviewing our work! We disagree that our analysis is "handwaving" and made an effort to clarify our exact contributions in the revised manuscript. We elaborate below.
>
>
> **Novelty:**
>
> > Many statements in Section 3 and 4 may be partly known in the literature, so they may not be novel.
>
> You question the novelty of our statements in Sec. 3 and 4. Please note that Sec. 3 is the ''Background'' section (and named as such) in which we summarize previous work necessary to understand our contribution. We include this material to make our submission self-contained. Therefore, lack of novelty is beside the point of this section.
>
> We are confident that our contributions in Sec. 4 are novel. If you are aware of work that already established any of them, we would be grateful for a reference!
>
> Our Sec. 4 develops the precise relation between noise-contrastive estimation (NCE) and negative sampling (NEG). As acknowledged in the Related Work section and at the beginning of Sec.4, some aspects of the relation between NCE and NEG have been discussed in Dyer 2014, Goldberg & Levy 2014, Levy & Goldberg 2014, and Ruder 2016. But to the best of our knowledge, these works either focused on the specific aspects of Mikolov et al. 2013's application of NEG to word embeddings, or considered an unrealistically high value for the number of noise samples $m$, or focused on an empirical comparison. The precise, theoretical connection we give in Cor. 2 is novel and crucial for understanding the exact relation of NEG to NCE. In particular, it applies in the realistic setting of small $m$.
>
> Prior to our work it was not known that NEG's minimum is a scaled version of the data distribution with proportionality factor $|X| / m$. For example, a recent survey on contrastive learning methods by Le-Khac et al. 2020 describes NEG as follows: "Also inspired by NCE, Mikolov et al. proposed a slightly different method called Negative Sampling (NEG) that focuses solely on learning good word representations with the trade-off of losing the probabilistic properties from NCE." However, as we showed, NEG does in fact learn the data distribution up to a known constant factor and can hence be used for learning language models and not just word representations.
>
> We have edited Sec. 4 and the Discussion to emphasize these points.
>
> **Formality:**
>
> > In my opinion, most arguments made in this paper are quite casual and handwaving.
>
> We are surprised about the criticism that our work is "casual" and "handwaving". Our key insight, Cor. 2, is presented as a formal statement with clearly specified assumptions and proof. In addition to this theoretical proof, we provided empirical evidence in Figs. 1, S10-S15, especially panel (i). Perhaps the large number of considered datasets has given the reviewer the impression that our work is only empirical. However, all these datasets only play a supporting role, whereas the central claim, Cor. 2, is mathematically precise.
>
> To further formalize our exposition, we have rephrased the central statement of Sec. 6 into a lemma, again giving a clear proof.

---

> > ### Author Response · Authors · 2022-11-16
> > **Reply to reviewer fZKZ (2/2)**
> >
> > **Discussion of ingredients of $t$-SNE and UMAP:**
> >
> > > There are also crucial parts of t-SNE and UMAP that are not carefully discussed. For example, it is known that using random initialization or spectral initialization has a great impact on the visualization quality.
> >
> > We do discuss all the ingredients to $t$-SNE and UMAP (as well as NCVis) in the second paragraph of the Related work section and in Sec. 3.2. We also explain why it is the loss that is responsible for characteristic $t$-SNE-like and UMAP-like appearances. In our experiments, we always use the same $k$NN graph, same Cauchy kernel, same initialization, etc. in order to study the influence of the loss function in isolation.
> >
> > In particular, we state that "Initialization was found to be strongly influencing the global structure in for both methods (Kobak & Linderman, 2021).", acknowledging the importance of initialization. Modern implementations of $t$-SNE such as Poli&ccaron;ar et al. 2019 use an informative PCA initialization by default, so that initialization has become less of an issue in practice. Moreover, we discuss that $t$-SNE and UMAP use different high-dimensional similarity distributions and different low-dimensional similarity kernels with slightly different tails by default. But, as we further elaborate in Sec. 3.2, these differences are largely inconsequential (Böhm et al. 2022 and Damrich et al. 2021). Even when controlling for all of them: Initialization with PCA, using the binary $k$NN graph as high-dimensional similarity and the Cauchy kernel as low-dimensional kernel in both $t$-SNE and UMAP, the characteristic discrepancy between the two embeddings persists. This discrepancy must therefore be the result of $t$-SNE's and UMAP's different loss functions. Consequently, our paper explores the relationship between their loss functions.
> >
> > Note that unless otherwise stated, we control for initialization, high-dimensional similarity, and low-dimensional kernel in all plots, including Fig. 1. The only exceptions are Figs. S17 and S18, where we ablate the choice of initialization and find NCVis and InfoNC-$t$-SNE to be more robust against poor initialization than $t$-SNE.

---

### Official Review · Reviewer_d2PD · 2022-10-24

**Confidence:** 2
**Correctness:** 2
**Technical Novelty And Significance:** 2
**Empirical Novelty And Significance:** 2
**Recommendation:** 6

**Clarity, Quality, Novelty And Reproducibility:**

The paper is clearly written, and the theoretical soundness is okay. Also, exploring the connection between two major dimensionality reduction algorithms is novel. Finally, the authors also provide their code as supplementary material.

In the meantime, the authors rely on the appendix too much, and interpretation of the experimental results lack in the main body. I suggest authors to add up more informative interpretation in the main body.


**Strength And Weaknesses:**

The paper extensively covers from neighbor embedding to self-supervised learning, and the authors linked those two concepts, which has the originality. Also, the paper is easy to follow, and the experimental results are promising.

However, at the same time, the quantitative result seems to be lacking. Are there any other quantitative result that can be shown in the experiment of dimensionality reduction algorithms?


**Summary Of The Paper:**

This paper relates tSNE and UMAP, starting from linking noise contrastive estimation and negative sampling. The authors also provided analysis on linking neighbor embedding and self-supervised learning, and it leads to optimizing tSNE with InfoNCE loss. The provided experimental results demonstrates the authors' arguments.

**Summary Of The Review:**

While I’m positive on this paper, I’m not very familiar with experiment parts of dimensionality reduction algorithms. Hence, I’m willing to see other reviewer’s comments.

---

> ### Author Response · Authors · 2022-11-17
> **Reply to reviewer d2PD**
>
> Dear reviewer d2PD,
> we thank you for the time and effort invested into reviewing our work and for your positive feedback!
>
> **Quantitative evaluation:**
> There are numerous quantitative metrics for dimensionality reduction. While none of them is perfect, they broadly fall into two classes: Metrics measuring local quality of the embedding and metrics measuring global structure preservation.
>
> Following your suggestion, we have conducted an extensive evaluation of the Neg-$t$-SNE spectra of all datasets in the paper using both a local metric, $k$NN recall, and a global metric, Spearman correlation between pairwise distances. The results can be found in appendix H (Figure S4).
>
> We found that:
>
> 1. Decreasing $\bar{Z}$ from UMAP's normalization constant to $t$-SNE's partition function improves the local faithfulness of the embedding while increasing $\bar{Z}$ leads to better preservation of the global structure, in line with our description of the local-global trade-off in the Discussion.
>
> 2. Neg-$t$-SNE embeddings align best with the proper $t$-SNE embedding when $\bar{Z}$ is close to $t$-SNE's partition function.
>
> 3. Neg-$t$-SNE embeddings align best with the proper UMAP embedding when $\bar{Z}$ is close to UMAP's normalization constant.
>
> Additionally, we computed both metrics also for the NCVis and InfoNC-$t$-SNE embeddings in Figs. S17 and S18, where we found that a higher number of noise samples improves the $k$NN recall, while random initialization deteriorates the Spearman correlation. We saw that both metrics became more similar to the results on $t$-SNE when increasing the number of noise samples $m$, underlining that NCVis and InfoNC-$t$-SNE approximate $t$-SNE better when using more noise samples.
>
> Altogether we believe that these experiments fully confirm our theoretical findings, and provide numerical evaluation of our claims. Thank you for suggesting this empirical support of our work!
>
> **Reliance on the appendix:**
> We do realize that our Appendix is rather extensive, and has now become even more extensive after we conducted additional experiments suggested by the reviewers! We have followed your advice and extended the interpretation of the results in the main text (see Discussion). We may further move some parts of the Appendix into the main text, but this will require moving some parts in the other direction. If there are some specific parts of the Appendix that you would prefer to see in the main text, please let us know!

---

### Author Response · Authors · 2022-11-22
**Novelty**

Dear reviewers, dear area chair,

some reviewers (gEqf and fZKZ) have doubted the novelty of our work.  We respectfully disagree. Reviewer fZKZ did this without pointing to any literature, making a rebuttal impossible. We maintain that our results are novel and briefly recap our main new insights here.

1. **Precise connection between NCE and NEG.**

NCE and NEG are popular contrastive learning methods used in many domains such as NLP ([Mnih et al. 2012](https://icml.cc/2012/papers/855.pdf), [Mikolov et al. 2013](https://proceedings.neurips.cc/paper/2013/file/9aa42b31882ec039965f3c4923ce901b-Paper.pdf), [Logeswaran et al. 2018](https://openreview.net/forum?id=rJvJXZb0W), [Kong et al. 2020](https://iclr.cc/virtual_2020/poster_Syx79eBKwr.html)), graph representation learning ([Grover et al. 2016](https://cs.stanford.edu/~jure/pubs/node2vec-kdd16.pdf)), neighbor embeddings ([Artemenkov et al. 2020](https://dl.acm.org/doi/fullHtml/10.1145/3366423.3380061), [Tang et al. 2016](https://dl.acm.org/doi/10.1145/2872427.2883041), [McInnes et al. 2018](https://arxiv.org/abs/1802.03426)), etc. However, their exact relationship remained unclear, and it is commonly believed that NEG is merely an _ad hoc_ non-probabilistic method ([Le-Khac et al. 2020](https://ieeexplore.ieee.org/document/9226466)). Here, for the first time we a give simple and precise probabilistic interpretation of NEG, emphasizing its difference from NCE. Our explanation of their connection can potentially benefit several fields listed above.

2. **Exact relationship between $t$-SNE and UMAP.**

$t$-SNE and UMAP are the two most popular non-linear dimensionality reduction algorithms, widely used across many application areas. They produce qualitatively different results, and the reason for this has remained unclear and much debated. Recent work (e.g. [Böhm et al. 2022](https://www.jmlr.org/papers/v23/21-0055.html)) tended to explain the difference by the fact that $t$-SNE is non-contrastive whereas UMAP is a contrastive method and only samples a small number of repulsive forces. However, $t$-SNE can also be optimized contrastively, e.g., using NCE. We use our insight into the NCE/NEG connection to give the first **conceptual** explanation of the relationship between $t$-SNE and UMAP.

3. **PyTorch framework for contrastive neighbor embeddings.**

We provide a swiss-army-knife modular PyTorch implementation of various contrastive neighbor embedding methods. It includes the first approximation of $t$-SNE with the InfoNCE loss, and the first parametric versions of NCVis and InfoNC-$t$-SNE. This serves three purposes: (1) it will help to switch between methods in practical applications; (2) it will allow ML researchers to investigate these different methods further in a unified fashion; (3) it underscores deep similarities between methods like $t$-SNE and self-supervised methods like SimCLR (which can also be optimized using our PyTorch module).

We are happy that the other three reviewers appreciated the novelty of our work (d2PD: "the connection between two major dimensionality reduction algorithms is novel", xsMD: "The paper [...] is very novel", ZbpN: "The paper [...] offers novel insights into the relation of NCE & NEG").

---

### Decision · Program_Chairs · 2023-01-20

**Decision:**

Accept: poster

**Justification For Why Not Higher Score:**

The paper provides an insight into t-SNE and UMAP that is likely interesting to the community. However, it does not rise to the level of a spotlight paper in terms of novelty and potential impact.

**Justification For Why Not Lower Score:**

The paper seems to provide an interesting insight to the community.

**Metareview: Summary, Strengths And Weaknesses:**

I would recommend accepting this paper for the following reasons:
- The paper investigates the connection of the two most popular contrastive learning methods between t-SNE and UMAP with interesting insights.
- The paper seems highly relevant to the community, as evidenced by two reviewers strongly championing the paper ("I use t-SNE and UMAP almost everyday. I always wonder what is the connection between them. This paper answers my question.").
- The author seem to adequately address reviewer concerns related to prior work.

**Note From Pc:**

if the above contains the word "oral" or "spotlight" please see: "oral" presentation means -> notable-top-5% and "spotlight" means -> notable-top-25%. As stated in our emails, we are disassociating presentation type from AC recommendations